# A Review of the Applications of Deep Learning-Based Emergent Communication

**Brendon Boldt**                                                    *bboldt@cs.cmu.edu*
*Language Technologies Institute*
*Carnegie Mellon University, Pittsburgh, PA, USA*

**David Mortensen**                                                  *dmortens@cs.cmu.edu*
*Language Technologies Institute*
*Carnegie Mellon University, Pittsburgh, PA, USA*

**Reviewed on OpenReview:** *https://openreview.net/forum?id=jesKcQxQ7j*

## Abstract

Emergent communication, or emergent language, is the field of research which studies how human language-like communication systems emerge *de novo* in deep multi-agent reinforcement learning environments. The possibilities of replicating the emergence of a complex behavior like language have strong intuitive appeal, yet it is necessary to complement this with clear notions of how such research can be applicable to other fields of science, technology, and engineering. This paper comprehensively reviews the applications of emergent communication research across machine learning, natural language processing, linguistics, and cognitive science. Each application is illustrated with a description of its scope, an explication of emergent communication's unique role in addressing it, a summary of the extant literature working towards the application, and brief recommendations for near-term research directions.

## 1 Introduction

Deep learning-based methods in natural language processing and multi-agent reinforcement learning provide a powerful way simulate how human language-like communication systems emerge *de novo*. This area of research is called *emergent communication* or *emergent language*. Multi-agent reinforcement learning-based systems like AlphaZero (Silver et al., 2017) and OpenAI's hide-and-seek agents (Baker et al., 2020) have leveraged self-play to exhibit convincing examples of complex behavior emerging from basic environment dynamics. Such deep reinforcement learning techniques were applied to discrete communication systems starting in 2016 and 2017 with papers like Foerster et al. (2016); Lazaridou et al. (2016); Havrylov & Titov (2017); Mordatch & Abbeel (2018). Although replicating as complex a behavior as human language is intuitively important, it is necessary to complement such notions with clear directives as to how it could apply to other areas of science, technology, and engineering.

Thus, this work is a review of the most salient goals and applications of deep learning-based emergent communication research. We illustrate each of the applications by providing a description of its scope, an explication of emergent communication's unique role in addressing it, a summary of the extant literature working towards the application, and brief recommendations for near-term research directions. This work has three primary goals. (1) This work is meant to inspire future emergent communication research by compiling the most salient areas of research into a single document with relevant work cited. (2) It illustrates to practitioners outside of emergent communication the potential ways that emergent communication can be used in an easily-referenced format. (3) It define the ultimate aims of emergent communication, which is critical to guiding the field of research through practices like establishing evaluation metrics and benchmarks. Evaluation metrics require explicitly defining what a *good* or *desirable* emergent language is, and understanding what emergent communication can be used for is a foundational step in their development.

Figure 1: Structure of the applications discussed in this review.

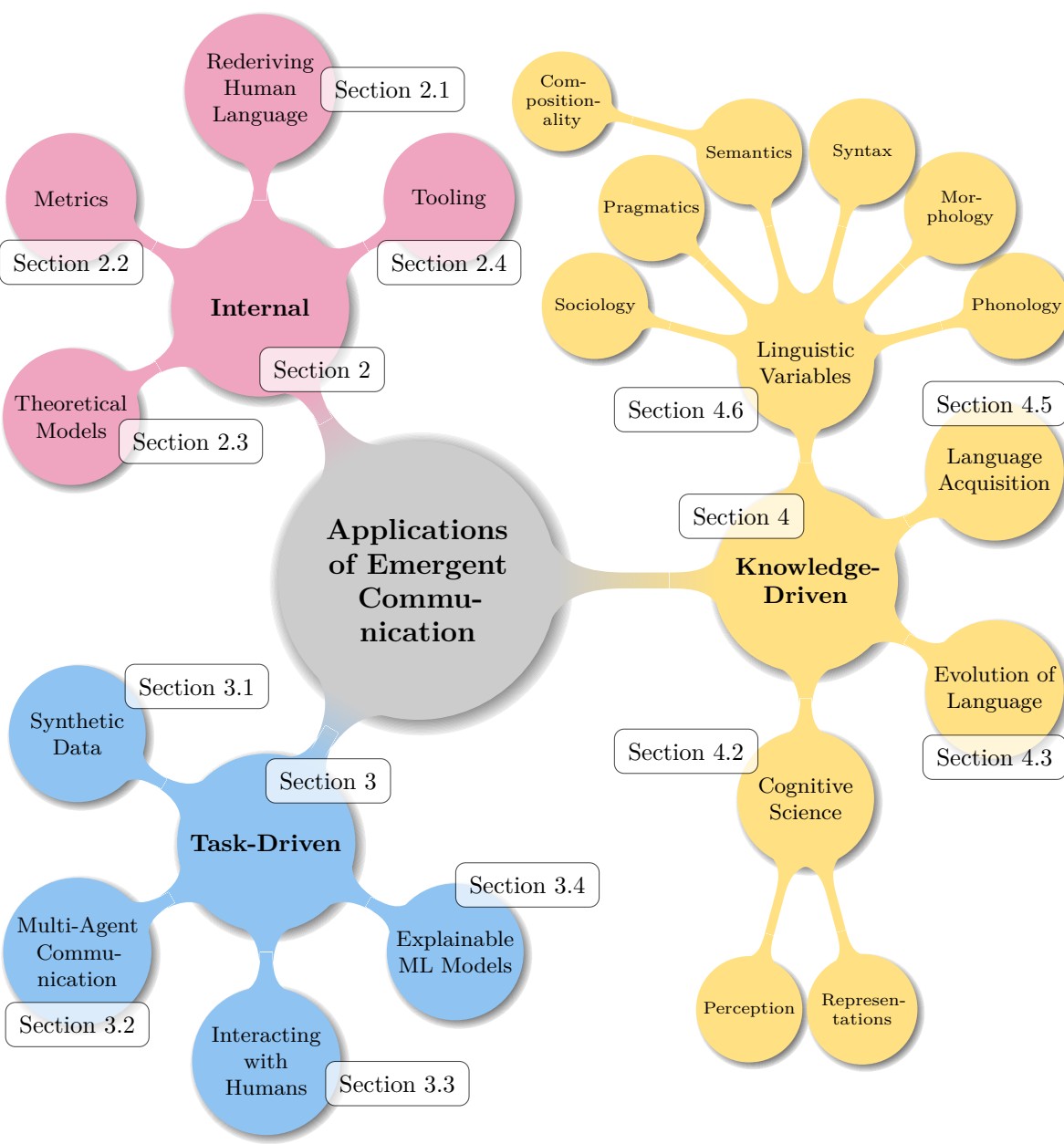

## Contents

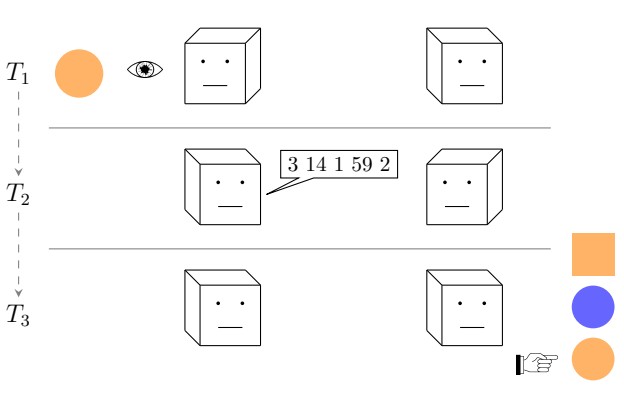

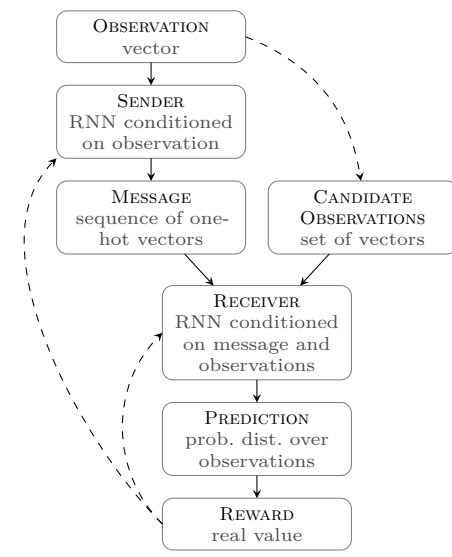

(a) $T_1$: The sender (left) observes an object. $T_2$: The sender passes a message to the receiver (right). $T_3$: The receiver chooses from a handful of candidate objects.

(b) Illustration of the technical architecture of the signaling game.

Figure 2: An illustration of the discrimination variant of the signaling game, one of the simplest and most common environments in emergent communication research.

## 1.1 Example of emergent communication system

In this section, we briefly illustrate a canonical example of an emergent communication game, namely discrimination variant of the signaling game (Lewis, 1969). The signaling game is one of the simplest and most common emergent communication games in the literature, and many further games and environments can be conceptualized as extensions of the signaling game. As seen in Figure 2a, the basic signaling game involves two agents, a sender and receiver. In a single round of the game, the sender first observes an object, then sends a message to the receiver, and finally the receiver chooses an object after observing a set of candidate objects along with the message from the sender. The round is successful if the receiver chooses the object which corresponds to the original observation made by the sender.

The technical architecture of the signaling game is illustrated in Figure 2b. The initial observation made by the sender is represented by a real-valued vector which which is an input for the neural network. The sender is a sequence generation model conditioned on the observation vector; RNNs are a common choice of architecture, but a number of other architectures can also be used. The sender generates a sequence of one-hot vectors which will serve as the "message" sent to the receiver. The receiver, typically an encoder RNN, then takes as input both the message and the set of candidate observations. The set of candidate observations contains both the correct observation made by the sender as well as "distractor" observations which differ from the correct one (i.e., like wrong answers on a multiple-choice question). Finally, the sender and receiver receive a reward based on whether or not the receiver selected the correct observation. This reward is then used to optimize the sender and receiver (e.g., with gradient descent[1]).

In the beginning, there is no pre-established communication protocol; that is, the messages produced by the sender do not "mean" anything. It is only through the repeated trials and optimization that messages begin to take on meaning such that the sender can effectively communicate the correct observation to the receiver. The protocol after training is considered the "emergent language" since it is a communication system which is the result of the functional pressure to succeed provided by the optimization of the sender and receiver.

---

[1]Since the message, which is a sequence of one-hot vectors, is discrete, it is typically required to optimize the sender with some additional technique like REINFORCE (Williams, 1992) or Gumbel-Softmax (Jang et al., 2017; Maddison et al., 2017).

### 1.2 Scope

In order to effectively select papers for the review, we need to define particular scope of "emergent communication" that we are dealing with. We are not claiming that work excluded by these criteria is unimportant or unrelated to the included work, nor are we arguing that these criteria should be viewed as normative. Rather, these criteria are merely intended to be sufficient for conducting a complete, coherent review of a field of research. The scope of this review specifically comprises the following criteria:

- *Necessarily*, the topic is an agent-based computer model, that is, the simulation of individual computer agents in an environment.
    - *Typically*, it uses reinforcement learning.
    - *Necessarily*, it is not simply the result of training a model on human language data (e.g., emergent properties of pre-trained language models do *not* qualify).
    - *Typically*, the system contains multiple agents (e.g., an agent talking to itself could still qualify).
- *Necessarily*, the agents have a communication channel.
    - *Necessarily*, the communication is analogous to human language in some way.
    - *Typically*, the communication channel is discrete symbols (i.e., analogous to words or subword units).
    - *Sometimes*, the communication channel may be continuous (e.g., analogous to speech sounds), but the structure of the channel or the resulting protocol must be of interest (i.e., an unconstrained, unstudied continuous channel does *not* count.)
- *Necessarily*, the exact nature of communication (e.g., the structure or content of the protocol) is not determined ahead of time; it "emerges" from simpler characteristics of the environments and agents.
- *Necessarily*, the approach uses deep learning methods.
    - *Typically*, methods use neural networks optimized by gradient descent.
    - *Typically*, work is associated with the communities of ICLR[2], NeurIPS[3], and ICML[4] conferences and the EmeCom workshop[5].

### 1.3 Related work

This section briefly discusses some closely related areas of research that fall outside of the scope of this paper. Although the goals and applications of these research areas are relevant to those discussed in this paper, we do not incorporate these into this paper in the interests of length. While their applications are very similar to those of deep learning-based emergent communication, the particular issues, methods, and possibilities which deep learning techniques present are quite different from these related areas.

**Emergent communication** Lazaridou & Baroni (2020) offer a general review of emergent communication research. It covers the same body of literature as this paper but with a general scope. Readers unfamiliar with the field of emergent communication would benefit greatly by reading this review first as it covers the essential elements of the field (background, methods, results, related work, etc.). This paper, on the other hand, focuses specifically on the goals and applications of emergent communication research.

A few other position papers have been published on emergent communication and share this paper's goal of guiding future work through a direct analysis and discussion of the literature. LaCroix (2019); Moulin-Frier & Oudeyer (2020); Galke et al. (2022) synthesize linguistic research on the evolution of language with contemporary methods in emergent communication, highlighting what aspects do not line up and how emergent communication research might change its approach. Finally, Zubek et al. (2023) provide a more robust critique of current methods in emergent communication from various linguistic perspectives.

---

[2]https://iclr.cc/
[3]https://neurips.cc/
[4]https://icml.cc/
[5]https://sites.google.com/view/emecom2022

**NLP and multi-agent RL**   Some areas of natural language processing focus on learning human language by leveraging deep multi-agent reinforcement learning in a way similar to emergent communication. This includes approaches like Lee et al. (2019); Cogswell et al. (2020) which use multi-agent dialog grounded with visual referents or Lu et al. (2020) which uses iterated learning framework for tuning dialog agents. Although the methods are similar, these approaches typically do not care about communication systems that are emerging from scratch and instead focus directly on improving performance with human languages.

**Emergent language without deep learning**   Computer simulations of the emergence of language are also possible without recourse to deep learning methods. Simulations along these lines might use other forms of machine learning or simply mathematical models of agents and environments. For example, Werner & Dyer (1991) simulate the emergence of a communication system in a population of mating animals. Female animals guide the male animals towards them by emitting discrete messages. Each agent is implemented as a connectionist artificial neural network which is optimized with a genetic algorithm. Another instance is Kirby (2000), which verifies the possibility of compositional communication emerging without biological evolution. Specifically, it presents a mathematical model of an agent population where new members must learn to communicate from older members (who eventually die off). This is implemented as a computer program which can empirically verify the hypothesis.

Although this research area has significant overlap in terms of goals, the methods have significantly different challenges. In particular, methods not based on deep learning tend to have strong inductive biases which constrain the range of languages that can emerge. In contrast, one of the main challenges of deep learning-based emergent communication is trying to find the environmental pressures and functional advantages which shape language in place alongside the weaker inductive biases of deep neural networks.

**Emergent communication with humans**   Research on the emergence of human language also takes place outside the context of computer simulation altogether. Experimentally, small-scale studies can be done with humans in the laboratory. For example, Kirby et al. (2008) test the emergence of structure in language from language transmission dynamics by having humans serve as the agents in a laboratory experiment. Observationally, there are recorded instances of a full human language emerging as in the case of Nicaraguan Sign Language, where deaf children with minimal prior linguistic knowledge developed language when placed together in a school environment (Kegl et al., 1999). Despite the relevance of this research to deep learning-based emergent communication, the challenges of human-base studies diverge significantly from those based on machine learning.

**Symbol emergence in robotics**   Taniguchi et al. (2015) survey a research area called "symbol emergence in robotics" (SER). SER is concerned with developing autonomous robots with the ability to discover meaning and communication skills from sensory-motor experiences with humans and other robots. In this way, both SER and emergent communication study "bottom-up" methods of autonomous agents acquiring the ability to use language in a deep, embodied way. SER is more concerned with the development of robotic agents which can dynamically learn to interact like humans through pragmatic and social facets of language. In contrast, emergent communication is more concerned with observing the entire process of language creation in virtual environments through agents interacting with other agents.

## 1.4   Structure of review

We divide the applications of emergent communication into three broad categories:

- *Internal goals* (Section 2) aim towards improving the technique in its own right. In a sense, these goals are the "basic research" of emergent communication.

- *Task-driven applications* (Section 3) aim at solving a well-defined problem. These goals generally correlate with goals in the domain of engineering such as those found in NLP.

- *Knowledge-driven applications* (Section 4) aim at increasing the knowledge of some phenomenon. These goals generally correlate with the goals of sciences such as linguistics.

We illustrate each application with four sections which each answer a question:

- "Description": What exactly is the problem being solved?

- "Applicability": How do the techniques of emergent communication (in practice or in theory) uniquely address this problem?

- "Current state": What progress has been made in the literature toward addressing this problem?

- "Next steps": What does the next important research paper towards this application look like?

We give a brief of analysis of the trends we found in reviewing the literature in Section 5 before concluding in Section 6. Details of the review process are presented in Appendix A, and a complete list of works surveyed is given in Appendix B.

## 2 Internal Goals

Internal goals are not what we would typically consider applications at all since they are focused on issues internal to the field of emergent communication. Nevertheless, these applications are important because they are (1) prerequisites for applying emergent communication to other areas and (2) the primary contributions of many papers that reference these goals. While, in a sense, any contribution could be considered an internal goal, we choose to address the internal goals which represent the clearest and most salient waypoints within emergent communication research.

### 2.1 Rederiving human language

**Description** Aiming to create emergent communication that resembles human language is a central characteristic of emergent communication and drives much of the research on the topic. This resemblance can include everything from low-level traits like compositional semantics and tree-like syntax to high-level traits like implicature (pragmatics) and sociolects (sociolinguistics), although how exactly to define this resemblance is an open question within linguistics. Aiming for resemblance does not necessitate exact replication of human language (or even having an exact definition of it): within human language we see a large amount of variation upon fundamental commonalities (e.g., distinguishing between nouns and verbs, having distinct units of meaning which appear in many contexts). *Rederiving human language*, then, is the process of developing the conditions (e.g., environment, agent architecture, games) which produce emergent communication which resembles human language.

Rederiving human language distinct from, although related to, *Origin of language* (Section 4.3) and *Language acquisition* (Section 4.5). Research into the origin and acquisition of language has a primary interest in the specific historical, environmental, and cognitive contexts of humans and their use of language. In contrast, rederivation is only concerned with these contexts for their instrumental value in developing emergent communication which is similar to human language.

**Applicability** The resemblance of emergent communication to human language is the nexus of most other goals in the field: other goals will either work toward resemblance in some respect or derive their effectiveness from resemblance to human language (or both). This resemblance to human language need not be perfect—even a partial rederivation of human language could still support many important downstream applications.

*Internal Goals* (Section 2) primarily work toward the rederivation of human language, while the task- and knowledge-based applications primarily derive their effectiveness from the rederivation. In particular, *Task-Driven Applications* (Section 3) rely on emergent communication having: structural similarities to human language (*Synthetic language data* (Section 3.1)), generalizability to new situations (*Multi-agent communication* (Section 3.2)), discourse structure (*Interacting with humans* (Section 3.3)), and the capacity to externally represent internal states (*Explainable machine learning models* (Section 3.4)). *Knowledge-Driven Applications* (Section 4), for example, rely on emergent communication resembling human language in terms

of: cognitive processes influencing linguistic behavior (*Language, cognition, and perception* (Section 4.2)), macro-scale social processes (*Origin of language* (Section 4.3) and *Language change* (Section 4.4)), mechanism of learning and acquisition (*Language acquisition* (Section 4.5)), and general structure at every level (*Linguistic variables* (Section 4.6)).

**Current state**   No work in the current body of literature has explicitly pursued the rederivation of human language. There are a large number of papers that study aspects of making emergent communication more human language-like in isolation (almost any paper in *Linguistic variables* (Section 4.6) does this in some capacity), but no papers have made steps towards rederivation holistically. While studying just one aspect of emergent communication at a time yields more tractable research questions, the risk is that isolating individual aspects misses the ways in which emergent communication is truly an *emergent* phenomenon within a complex system (Bar-Yam, 2002, Sec. 1.3). Complex systems are characterized by non-obvious interactions among the many moving parts, and taking away single elements of the system might change the behavior in significant, unpredictable ways. To the extent to which this is true, studying isolated phenomena in simple environments has limited potential.

For example, Ren et al. (2020) show, in line with established experiments with mathematical models (Kirby et al., 2007) and human subjects (Kirby et al., 2008), that the imperfect transmission of language from generation to generation (i.e., *iterated learning*) can explain a bias toward compositionality in communication system without further agent-internal biases. Yet empirical investigation of compositionality in emergent communication literature often uses fixed-population environments[6]. The fact that iterated learning has diverse support as an explanation for compositionality calls into the question the results of compositionality research which does *not* take iterated learning into account, since iterated learning could be a sufficient driver for compositionality in emergent communication, outweighing other potential sources like model capacity (Resnick et al., 2020) or perception (Lazaridou et al., 2018).

**Next steps**   The rederivation of human language in full is a massively complex task which may be impossible in practice or even in principle. Yet even if it is possible only to a limited degree, emergent communication can still fulfill many of its applications. The first step toward rederiving human language is laying down the theoretical foundations: identifying the most salient properties of human language and using these to develop a concrete problem definition of "rederiving human language". The field of linguistics will be especially important for formulating precise notions of "rederiving human language". Such formulations will provide the groundwork for identifying the technical issues with rederiving human language through emergent communication techniques. For example, we speculate that: language will need to be processed by larger neural networks with parameter counts in the billions; agents will also need to have realistic cognitive constraints on producing and understanding language; populations of agents will have to number in the hundreds to mimic even the smallest human language communities; environments will need to be scaled up in terms of both sensory input (e.g., 3-dimensional environments, embodiment) as well as task complexity (e.g., involving multi-step planning); and many advanced techniques from deep reinforcement learning will need to incorporated into the optimization process in order to learn from richer environments (e.g., efficiently learning representations, planning, multi-agent cooperation).

## 2.2   Metrics for emergent communication

**Description**   A metric, for our purposes, is a well-defined method for quantifying a property of or notion about an emergent communication system. Some properties in emergent communication are fairly concrete and are naturally quantitative such as vocabulary size or task success rate. Other properties are more abstract and there is not single, obvious way to quantify them (i.e., they are underspecified in some capacity). For example, compositionality often refers to the idea that "the meaning of a composite message is a function of the meanings of individual parts", but this definition is underspecified. It does not specify if "meaning" rests in the interpretation of the speaker, listener, or both, nor does it specify what limits might exist on functions used to combine meaning—each interpretation would be quantified differently and may be useful in different

---

[6]I.e., environments where the set of agents remains constant throughout the training process. Contrast this with dynamic populations where newly-initialized agents enter the population and older agents leave the population.

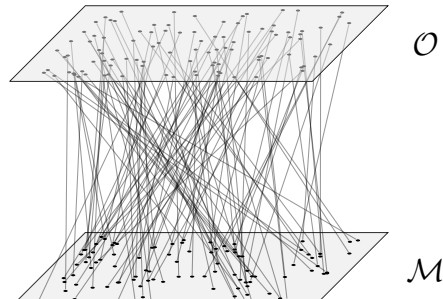 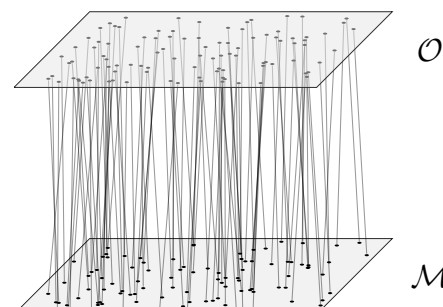

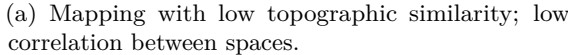

(a) Mapping with low topographic similarity; low correlation between spaces.

(b) Mapping with high topographic similarity; high correlation between spaces.

Figure 3: Illustration of two spaces with different topographic similarities (*toposims*). $\mathcal{O}$ and $\mathcal{M}$ represent embedding spaces for the observations and messages, respectively. A high toposim means that distances between any two points in the observation space correlates well with distances between corresponding points in the message space.

contexts. Finally, *evaluation* metrics are even more abstract as they try to directly measure how *good*, *useful*, or *desirable* something is in a general sense. For example, F-score is an evaluation metric for classifiers; that is, a better classifier should have a higher F-score (insofar as F-score is an effective evaluation metric), and generally speaking, a classifier with higher F-score will be more useful than one with a lower F-score.

Thus, developing metrics within emergent communication comprises a number of different tasks, including: designing precise formulations of abstract properties, developing practical computational methods for implementing these formulations, and demonstrating mathematically and empirically that they accurately quantify the particular property.

**Applicability**   Metrics, in general, are a ubiquitous part of research in most any area of science or engineering. They are integral to formulating testable hypotheses since they delineate precisely what is being considered empirically (or theoretically). They are also what enables effective summarization and statistical analysis of the results of experiments beyond mere qualitative analysis. Together, these two factors make principled comparison with prior work possible. Evaluation metrics, in particular, help identify approaches to a given problem are most effective. These are especially important for the long-term development of a field as they help gauge overall progress and direct efforts towards the most promising approaches.

**Current state (compositionality)**   Metrics for compositionality and generalizability comprise the lion's share of literature on this goal while only a few have been developed for other properties. This corresponds with the most common goals of emergent communication papers which are to develop emergent communication which has compositional semantics and generalizes beyond the scenarios seen during training.

Compositionality (or compositional semantics) refers to the general principal that utterances with complex meaning derive their meaning from a combination of the meaning of the components of the utterance (e.g., a "red car" is a car that is red). This is in contrast to "holistic" communication where there is no relationship between the meaning of an utterance and its components.[7] The most popular metric for compositionality is topographic similarity (Brighton & Kirby, 2006; Lazaridou et al., 2018), which quantifies compositionality as the degree of correlation between distances in the referent feature space and distances in the message space (illustrated in Figure 3). Specifically, Lazaridou et al. (2018) use the Spearman's rank correlation coefficient ($\rho$) on the pairwise distances between objects in the feature space (quantified with cosine similarity) and their corresponding messages (quantified with Levenshtein distance). In this sense, toposim is more of a family of metrics since the precise methods of computing correlation and distance in the object and message spaces have a number of concrete realizations.

---

[7]For example, a "black swan" can refer (idiomatically) to a rare event—something that is neither black nor a swan.

Representation similarity analysis (Kriegeskorte et al., 2008; Luna et al., 2020) takes a similar approach to quantifying compositionality but measures the correlation in the feature space and agents' internal representations. A handful of other metrics fall under the umbrella of *disentanglement*, where components of the message specify single attributes and do so independent of context. Such metrics include positional and bag-of-words disentanglement (Chaabouni et al., 2020), context independence (Bogin et al., 2018), and conflict count (Kuciński et al., 2020b). Tree reconstruction error takes a deeper look at the compositionality of language by measuring how closely an explicitly compositional model of semantics can approximate what the emergent communication agents produce (Andreas, 2019).

In response to the amount of research into measuring compositionality, some papers have provided deeper analyses of metrics of compositionality. Korbak et al. (2020) provide a meta-analysis of the above compositionality metrics, showing that while many are sensitive to basic types of compositionality, most fail to recognize more sophisticated methods of composition. Kharitonov & Baroni (2020b); Chaabouni et al. (2020) provide evidence against the claims that standard measures of compositionality are also measuring the ability of the language to generalize to describing novel objects.

**Current state (other)**   Generalizability, as in other areas of machine learning, generally refers to the ability to perform well outside of the training conditions. Generalizability is most often operationalized as agents successfully describing objects previously unseen combinations of attributes (i.e., generalizing from training data to test data) (Korbak et al., 2019; Chaabouni et al., 2020; Denamganaï & Walker, 2020a; Kharitonov & Baroni, 2020a; Resnick et al., 2020; Perkins, 2021a). Apart from this type of generalization, other work has looked at generalizing to new communication partners (Bullard et al., 2021), generalizing over different environments (Guo et al., 2021; Mu & Goodman, 2021), and generalizing across linguistic structures (e.g., disentangled syntax and semantics) (Baroni, 2020).

Yao et al. (2022) introduce an evaluation metric, that is, one which measures the *overall* quality of an emergent language using data-driven methods. The metric equates the quality of an emergent language with the quality of machine translation from the emergent language to human language. The underlying intuition here is that the more human-like an emergent language is, the more effective substituting it for human language will be in machine learning tasks (i.e., using it as synthetic data, see Section 3.1).

**Next steps**   With regard to metrics of phenomena like compositionality, it is critical to incorporate knowledge from linguistics as to how similar notions apply to human language. For example, with compositionality, emergent communication research often use to simple notions of compositionality, focusing individual units of meaning combining arbitrarily to form composite meanings. On the other hand, human language's relationship with compositionality is far more complicated, sometimes exhibiting it while sometimes being non-compositional (e.g., idioms, irregular word forms, grammatical rules limiting "acceptable" sentences). While this cross-disciplinary approach more difficult to incorporate into the research process, it is critical to the long-term trajectory of emergent communication research.

Evaluation metrics, on the other hand, are mostly absent in the emergent communication literature despite their importance to other fields of machine learning like reinforcement learning and natural language processing. Thus, it would be fruitful to develop true evaluation metrics which can determine what emergent languages are "best" or "most human language-like". As these notions are more abstract than "compositionality" or "generalizability", there is more theoretical groundwork that must go with the motivation of evaluation metrics in addition to the engineering efforts in actually designing and implementing them.

### 2.3   Theoretical models

**Description**   A theoretical model of emergent communication is a mathematical or formal system which describes the behavior of an emergent communication system. Generally speaking, a theoretical model will describe a relationship between two or more variables in an emergent communication system. Theoretical models are developed in conjunction with empirical work and represent a refinement and systematization of the knowledge gained from these experiments. Most importantly, their formal representation allows rigorously reasoning about the behavior of a systems without needing to directly run experiments.

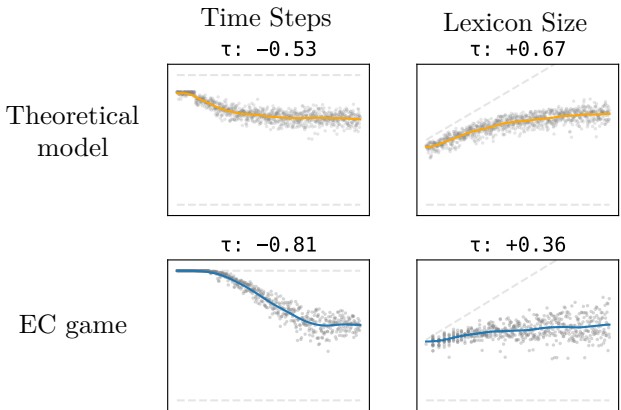

Figure 4: Plots of lexicon entropy ($y$-axis) versus time steps and lexicon size ($x$-axes) comparing a theoretical model against empirical measurements on navigation game with emergent communication (from Boldt & Mortensen (2022b)).[8] Theoretical models can help predict the outcomes of emergent communication games much more efficiently than simply running the environment while also providing a conceptual understanding the environment's behavior.

**Applicability**  Theoretical models benefit emergent communication research primarily in two ways: they clarify research methods and can predict a system's behavior in compute-intensive situations. For research methods, using a theoretical model to phrase a research question results in a hypothesis which is clear and testable. As a result, the empirical evaluation has a clear relationship with the assumptions and structure of the model, allowing subsequent research to more easily build on previous work. In the absence of theoretical models and their hypotheses, papers must often rely on qualitative hypotheses which are difficult to empirically verify or result in merely pointing out "interesting" observations from the experiments. For this reason, employing theoretical models can move emergent communication research towards systematically scientific investigation instead of less organized trial-and-error.

Second, theoretical models provide a way to predict the behavior of emergent communication systems in situations where directly running the system is computationally expensive. The applicability of theoretical models on this front is discussed in the context of GPT-4 (OpenAI, 2023) and its scaling laws where the extreme computational cost of training the model made it critical that the designers could predict the behavior of the full-scale model ahead of time. In particular, they fit a mathematical equation predicting the loss based on computational input using smaller models. This gave the developers a way to accurately predict the final loss of the full-scale model at a fraction of the computational cost. As emergent communication environments get more complex with more design choices, hyperparameters, and computational cost, it will also be important to be able to predict the behavior of the system without having to run the full environment in every situation.

**Current state**  Only a handful of papers in the literature use theoretical models, and these models are usually not employed in any subsequent papers. Khomtchouk & Sudhakaran (2018) study the transition between two degenerate "phases" of language: single-symbol systems and full one-to-one systems, with the synonymy and ambiguity found in human language lying in the middle. This model is then tested with a pair of simple reinforcement learning-based agents. Ren et al. (2020) apply the iterated learning model (Smith et al., 2003) to deep learning-based emergent communication; they use a formal probabilistic model of an iterated learning algorithm to express hypotheses which are empirically tested. Boldt & Mortensen (2022b) formulate a stochastic process which describes the entropy of an emergent language's lexicon based on a handful of hyperparameters of the agent's neural network; the predictions of this model are also empirically tested in four simple emergent communication environments. Rita et al. (2022b) analyze emergent communication environments based on the Lewis signalling game by providing a mathematical decomposition

---

[8]$\tau$ is the Kendall correlation coefficient of the points.

of the loss function. This decomposition explains the different overfitting pressures hidden in the loss function; from this, they suggest measures to counteract such pressures which result in more compositional emergent communication.

Finally, the model presented in Resnick et al. (2020) is a good representative of theoretical models in emergent communication and their attendant difficulties. The model describes the relationship between the capacity of an agent's neural networks and the compositionality of the learned emergent language: if the capacity is too low to capture the regularities in the language (i.e., grammatical rules) the agents "underfit", and if the capacity is too high, the agents "overfit" by simply memorizing individual utterances in the language. The model predicts the compositionality to be low in both the under- and overfitting regimes and higher between them where the neural network learns regularities without memorizing individual examples. The model could then be formalized as follows

$$\text{Capacity}(M_X) \in [C_L, C_U) \wedge \text{Capacity}(M_Y) \notin [C_L, C_U) \Rightarrow \text{Comp}(M_X) > \text{Comp}(M_Y) \tag{1}$$

where $M_X$ and $M_Y$ are the agents' underlying models, $\text{Capacity}(\cdot)$ quantifies a model's capacity, $C_L$ and $C_U$ are the under- and overfitting thresholds respectively, and $\text{Comp}(\cdot)$ quantifies the compositionality of the model's emergent language.[9]

The precise formulation of the model results in a clear hypothesis based on the predictions of the model, allowing the experiments to more directly test the underlying principles of the model. The difficulties that persist, though, are that the model's formulation and its predictions still lack precision. In the formulation, Resnick et al. (2020) do not fully articulate what constitutes "capacity"; a notion of capacity though would be extremely difficult if not impossible to formulate precisely for deep learning models. In the predictions, the paper is only able to articulate general trends and correlations rather than predicting exact values or distributions. These issues, though, are representative of more general issues with theoretical models in emergent communication the use of deep neural networks and reinforcement learning make precision inherently difficult (although approximation is not impossible as shown by the GPT-4 scaling case above). Finally, despite the fact that the model in Resnick et al. (2020) addresses compositionality, the most popular topic within emergent communication, it does not see reuse subsequent papers.

**Next steps**  Theoretical models are difficult to apply deep learning-based emergent communication since deep neural networks themselves are difficult to formalize. Part of this difficulty is inherent while some of it stems from the sparsity of formalization in the applications of deep neural networks. Thus, an important next step would be to address these difficulties with archetypical examples of how theoretical models can be applied to emergent communication as well as recommendations for best practices, taking inspiration from existing work on the theoretical foundations of deep learning and complex systems.

Even given these difficulties, one-off instances of simple theoretical models can be helpful in clarifying the contributions, hypotheses, and results of a given paper. For example, instead of hypothesizing simply that changing $X$ will improve $Y$, it could be stated instead that there will be a positive correlation between $x$ and $y$, where $x$ and $y$ are quantitative metrics of $X$ and $Y$ respectively, and correlation is mathematically defined (e.g., Spearman's rank correlation coefficient). This would facilitate experiments which more clearly refute or a support a hypothesis and its underlying claims.

## 2.4   Tooling

**Description**  The central aim of tooling within emergent communication is to develop apparatus that can be used to ease the process of implementing and running experiments. Since emergent communication is under the broad category of computer science, the experimental apparatus are most often programs, their source code, and sometimes datasets. Although any codebase used for an emergent communication experiment can be reused and repurposed by other researchers for new experiments, codebases which are designed to be reused for a broad range of experiments are the focus of this application.

---

[9]This is a summarization of the model which is more precise in its original formulation. The particular formalization is not used in the original paper and is instead derived from Boldt & Mortensen (2022c).

**Applicability**   The most obvious benefit of shared and standardized tooling is that it saves time for researchers as less time needs to be spent reimplementing the basic features of emergent communication experiments. Furthermore, the incidence of bugs decreases, implementation efforts can be spent improving existing tooling, and comparison across papers is more reliable since more implementation details will be the same. Special care must be taken, though, that the implementation details do not lead to systematic biases in experiments; emergent communication is especially susceptible to this concern since it is difficult to distinguish between effects of structure of the environment (abstractly speaking) and effects of implementation details. Finally, well-designed and easy-to-use tooling is a significant help to emergent communication researchers who do not have a strong software development background. The task of putting ideas into code is much more difficult for such researchers, and decreasing the amount of unnecessary reimplementation can greatly improve their ability to contribute to the field of emergent communication.

**Current state**   Tooling for emergent communication has a small degree of standardization, although the high degree of variety in problems and approaches in the field decrease the practicality of a one-size-fits-all framework. EGG (Emergence of lanGuage in Games) (Kharitonov et al., 2019) is the most widely used framework for deep learning-based emergent communication; it provides a simple Python programming interface for some of the most common emergent communication games, agent architectures, and metrics. Papers which implement new games using EGG further expand the range of games and metrics which are easily accessible through the framework including: Chaabouni et al. (2019b); Dessì et al. (2019); Chaabouni et al. (2020); Auersperger & Pecina (2022); Kharitonov et al. (2020b). ReferentialGym (Denamganaï & Walker, 2020b) is similar to EGG in scope, although it has seen less reuse within the literature. Other tooling may target more specific aspects of experiments in emergent communication. For example, TexRel (Perkins, 2021b) is a synthetic dataset designed specifically for use in emergent communication games; in this case, data (images) are constructed such that *compositional* language could aptly describe them. Additionally, Ikram et al. (2021) introduce HexaJungle, a suite of environments for studying emergent communication. For papers which explore beyond the typical environments (which is a significant portion), it is common to implement the emergent communication game and agents from scratch using more general purpose tools like PyTorch (Paszke et al., 2019) (Evtimova et al., 2017; Mu & Goodman, 2021; Noukhovitch et al., 2021).

**Next steps**   The next steps for tooling in emergent communication largely depends on what tasks, problems, and methods receive the most attention going forward. If the field continues to study similar environments, EGG could continue to support such work, but if radically new environments or experimental paradigms appear, current tooling might prove insufficient. In part, this is due to an inherent trade-off between the flexibility of a framework and its convenience; while emergent communication is rapidly changing, the required flexibility often does not provide much convenience, but as the field focuses on fewer problems, frameworks could play a greater role. A possible middle way between these two issues would be developing an interface (in the sense of object-oriented programming) for emergent communication environments similar to OpenAI Gym (Brockman et al., 2016), which can provide some standardization and interoperability while not impeding novel environments and implementations.

## 3   Task-Driven Applications

The task-driven applications of emergent communication center around fields of engineering such as machine learning, natural language processing, and multi-agent systems, and typically involve solving a well-defined, practical problems. These applications have the most immediate impacts and, as such, offer some of the most convincing motivations for developing emergent communication techniques in the short term. The primary challenges in this area come from competing against more established methods in deep learning which are continually advancing through larger and larger scales of data and compute.

### 3.1   Synthetic language data

**Description**   In the context of deep learning, synthetic data refers to data which is generated with a computer program; this is in contrast to "real" or "natural" data which is collected from an actual system being studied.  For example, the language we find in books, conversations, speeches, etc. would all be

real, in this sense, whereas a corpus of sentences generated by sampling from a probabilistic context-free grammar would be synthetic. For example, within NLP, synthetic data can be used for transfer learning (Papadimitriou & Jurafsky, 2020; Mirzaee & Kordjamshidi, 2022) and model probing (Lake & Baroni, 2018; Zhang et al., 2022). Synthetic data has a number of advantages when applied to deep learning; this includes: controllability, availability in arbitrary quantities, availability in low-resource domains (e.g., endangered languages, multi-modal settings), alleviating concerns about bias, and alleviating privacy concerns (since data is not collected from humans, e.g., through surveillance). Although synthetic data finds niche uses alongside real data in deep learning and NLP, it fails to have widespread applicability because it often does not capture the plethora of nuances and irregularities that appear in real data, that is, the "long tail" of the real data distribution. In natural language, this can manifest as unnatural but valid syntactic structures, uncommon senses of words, idioms, and wordplay.

We consider all kinds of pretraining, data augmentation, analyses, and evaluation of deep learning models with emergent communication as part of this application even if it does not involve generating synthetic datasets *per se*. For example, you might use a trained emergent communication agent itself to pretrain or evaluate a model instead of generating an intermediate dataset.

**Applicability**  Emergent communication could serve as a way to generate synthetic language data which more closely mimics the natural variation found in human language. The distribution of patterns within natural language has a "long tail" insofar as a large proportion of the total mass comprises a large number of infrequent patterns, making it very difficult for something like synthetic data generated by handcrafted programs to sufficiently replicate the distribution (Naik, 2022). This is illustrated by the history of NLP: handcrafted expert systems have been surpassed by learning-based method which can scalably leverage computing power to mine patterns from increasingly large quantities of data. Emergent communication, rather than mining patterns directly from data, seeks to uncover linguistic and behavioral patterns which are latent in the communicative pressures of embodied multi-agent environments. Work such as Artetxe et al. (2020) demonstrates that deep neural networks do learn latent, language-agnostic patterns from their training data; this suggests that even if an emergent language does not have a one-to-one correspondence with some particular human language, having underlying structural similarities with human language would be sufficient to still be useful.

**Current state**  Work towards using emergent communication to generate synthetic data has been at the proof-of-concept level. The papers in our survey (discussed below) showed that emergent communication could indeed improve the performance of neural NLP models when used for pretraining in very low-resource settings. That being said, experiments only cover a narrow selection of datasets/tasks and do not rigorously compare against alternative methods (e.g., traditional synthetic data, cross-lingual transfer). As a result, is difficult to gauge the practical impact of the proposed methods.

Li et al. (2020) pretrain encoder-decoder few-shot machine translation models with an emergent communication signalling game; in addition to finding improvements in very low-resource settings, the experiments showed that the task success rate in the emergent communication game was well-correlated with the downstream BLEU score. Downey et al. (2022) also tackle machine translation, but instead use an emergent communication game to fine tune a multi-modal model for unsupervised machine translation, finding that emergent communication is more effective than the back-translation baseline. Yao et al. (2022) take a slightly different approach by using the emergent communication game only to generate a synthetic corpus (instead of training the models directly); this corpus is then used to pretrain models for language modeling and image captioning tasks. The experiments compare emergent language corpora against two baselines: Spanish and a synthetic dataset generate by sampling delimiters from a Zipfian distribution to create a hierarchical language with similar structural biases to human language (e.g., {<()>[]()}). For the lowest data regimes, pretraining the model on emergent language corpora reliably outperforms models pretrained on the baseline datasets. Finally, Mu et al. (2023) use emergent communication to pretrain an instruction-following embodied control model (e.g., for controlling a robotic arm); the experiments showed that not only does the proposed method outperform the baseline models but also that the emergent language is more effective as training data than pre-trained, static representations derived from video demonstrations.

**Next steps** The first direction is thoroughly investigating the different ways emergent communication can be used for generating synthetic data. Li et al. (2020) (using emergent communication agent models directly downstream) and Yao et al. (2022) (using emergent language corpora for pretraining downstream models) take different approaches to the same task of pretraining downstream NLP models. These approaches have different relative merits (e.g., making better use of training data versus decoupling agent architecture from downstream architecture, respectively), and there are likely more ways to approach the same problem with emergent communication. Thus, next steps would consist of finding other promising methods of harnessing emergent communication for model pretraining and comparing these approaches on a common ground. Determining which of the approaches is best is critical to giving emergent communication the best chance of surpassing more traditional methods model pretraining and generating synthetic data.

The second direction which can be pursed after or in parallel to the first is rigorously comparing emergent communication for pretraining neural NLP models with more established techniques like cross-lingual transfer and traditional synthetic data (Artetxe et al., 2020). First and foremost, this helps to establish whether or not emergent language data can truly surpass what is already present in the field. In particular, comparison against cross-lingual transfer should highlight how emergent language data is more available, that is, it can be attained in higher quantities with more relevance to the target language than cross-lingual data. Comparison against traditional synthetic data could tease out exactly what properties of emergent communication make it more effective in downstream applications. For example, emergent communication could be compared against increasingly complex synthetic languages: balanced parentheses, context-free grammars, then full-scale grammars (e.g., head-driven phrase structure grammar (Pollard & Sag, 1994)).

Both directions would entail developing a sort of benchmark for testing the effectiveness of pretraining methods. This would require not only finding suitable data sources and evaluation metrics, as usual, but also determining how to make the variety of methods for pretraining comparable. For example, emergent communication is more computationally expensive than traditional synthetic data and standard NLP pretraining methods, yet it could surpass synthetic data in quality and real data in low-resource settings. Therefore the benchmark would have to take into account data and computational requirements in addition to raw performance.

## 3.2 Multi-agent communication

**Description** The area of multi-agent communication is concerned with autonomous (computer) agents coordinating their actions through the use of a communication protocol. Most prototypically, this would apply to a team of autonomous robots working together but could also include situations like self-driving cars on the road (illustrated in Figure 5) or IoT devices on a local area network. The two typical approaches to developing multi-agent communication protocols are handcrafting them or learning them like a latent variable between agents. Handcrafted protocols (e.g., DHCP for network configuration) are typically well-suited for specific tasks but are also require significant expert design, which hinders much potential for open-domain or general purpose communication. Automatically learned continuous protocols (i.e., messages are learned continuous vectors) solve some of these issues but raise new issues related to deep, learned representations such as low interpretability. This task is distinct from autonomous agents communicating directly with humans, which we discuss in Section 3.3.

**Applicability** Emergent communication addresses these issues in three main ways. First, emergent communication is scalable to more general-purpose tasks since it is developed by computational processes directly from the functional pressures of the task it is applied to. Second, it is more interpretable insofar as it resembles the structure of human language, for example, using discrete symbols in its communication channel or having a hierarchical syntactic structure (cf. continuous vectors which do not resemble human language and require mathematical transformation to be analyzed). Finally, human language is the gold standard for communication protocols insofar as it can apply to previously unseen situations and is robust to noise and other hindering factors. Thus, developing communication protocols which deliberately mimic the structural properties of human language could be a way to better attain these desirable functional properties.

For example, the following design elements of an emergent communication system could contribute to recreating some of the above desirable properties of an emergent language. To encourage general purpose language, we can start with an open world, open-ended environment (e.g., Minecraft) and/or one with many

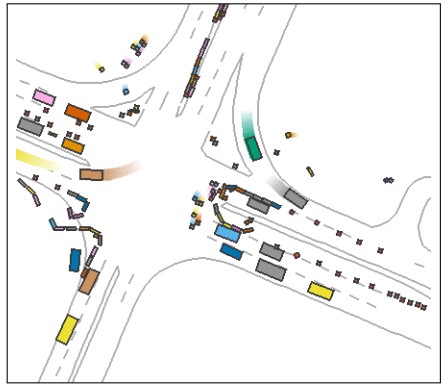
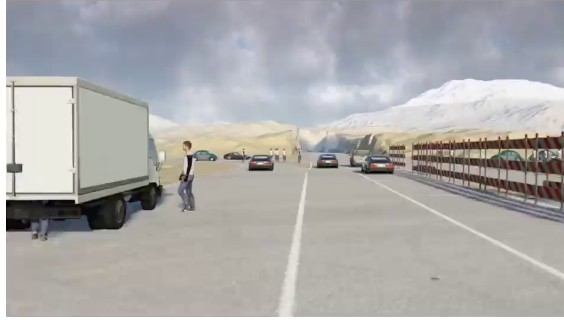

(a) Complex scenario with various kinds of agents.  (b) Pixel-based input to approximate real-world diversity in observations.

Figure 5: Self-driving vehicles in complex traffic situations are an important application of multi-agent communication. Diversity in both scenarios as well as the observations themselves indicate that open-ended communication systems could be more appropriate than handcrafted protocols where all scenarios are anticipated ahead of time. Screenshots from documentation of MetaDrive (Li et al., 2023) (Apache-2.0 license).

distinct situations (e.g., Starcraft, Dota 2). Furthermore, tasks which have adversarial components can especially elicit a diversity of situations since one team of agents is constantly trying innovate to outcompete the other. Towards interpretability, the agents could be constrained to communicate only with discrete symbols at human-scales (e.g., modest vocabulary size and message length). Finally, elements like communication channel noise or constantly cycling out agents in the population can induce a more robust communication protocol since agents cannot as easily overfit to each other.

**Current state** Work on developing multi-agent communication protocols has experimented with a handful of environments and scenarios but has not established any one task as being definitively helped by emergent communication. Many of the explored environments are a variation on navigation (Mul et al., 2019; Li et al., 2022; Masquil et al., 2022) or the signalling game (Bullard et al., 2021; Cope & Schoots, 2021; Wang et al., 2022; Tucker et al., 2021), although some include more abstract environments like a coalition-based voting game (Li et al., 2022) or semantic communication (Thomas & Saad, 2022). Emergent communication for multi-agent communication has been compared against competing methods, that is, handcrafted protocols (Gupta et al., 2020; Chen et al., 2022) and learned continuous communication (Li et al., 2022; Wang et al., 2022). Although, these comparisons use the competing methods more as "baseline points of reference" rather than comparing them "head-to-head", where both methods are presented in their strongest forms so as to show the real-world superiority of emergent language-based communication. In most cases, increasing the performance of the multi-agent team is the primary interest of the experiments; additionally, papers have also looked at emergent communication's robustness to corruption and noise (Cope & Schoots, 2021; Wang et al., 2022) as well as the potential for communicating with partners not seen during training (Bullard et al., 2021; Cope & Schoots, 2021).

**Next steps** The first direction of future work on emergent communication for multi-agent communication is to find a niche for emergent communication, that is, presenting a particular task where, in realistic conditions, emergent communication surpasses state of the art non-emergent approaches. Although emergent communication has intuitive advantages (discussed above in "Applicability"), it has yet to be shown in a real-world task. This is a significantly more difficult task than what most of the current literature accomplishes: namely demonstrating a on small scale that multi-agent communication is possible with emergent communication techniques as a proof of concept. Based on the particular advantages of emergent communication, such a task will likely have to be open-domain or demand continual adaptation, rendering hand-crafted protocols impractical, while also needing some element of interpretability, demonstrating an

advantage over learned continuous communication. This is a formidable task as presenting effective definitions of "open-domain" and "interpretable" require formalizing rather abstract notions.

In conjunction with this first direction, it will also be necessary to empirically verify the intuitions that (1) emergent communication is more interpretable than continuous communication, and (2) emergent communication's structural similarities to human language confer some actual functional benefit beyond continuous or unconstrained communication. If these intuitions are well-founded, then it will greatly expand the potential applications of emergent communication in multi-agent systems.

### 3.3 Interacting with humans

**Description**   A perennial goal of computer systems has been more naturally interacting and communicating with humans. This is an incredibly difficult task due to the complexities of human communication ranging from nuanced syntax and semantics to pragmatics and conversational dynamics. While deep learning methods have had good success learning syntax and decent success learning semantics, proficiency at the level of pragmatics is not yet present because these higher levels of language and communication tend to be more difficult to learn from purely from text through a language modeling objective. This is demonstrated in Ouyang et al. (2022) by the fact that an InstructGPT model outperforms a GPT-3 model $100\times$ larger when it comes to following a human's instructions (as evaluated by humans). They observe from this that the language modeling objective alone is misaligned with the objective of "follow the user's instructions helpfully and safely"; for example, truthfulness is one dimension that is drastically increased by training on human feedback (Ouyang et al., 2022). Even with extensive training with human feedback, models like ChatGPT still significantly diverge from humans when it comes to pragmatics and communication strategies (Qiu et al., 2023; Guo et al., 2023). Thus, despite large language models' fluency, they do not naturally capture critical aspects of interacting with humans, and current methods of addressing it entail relying directly on human supervision (Ouyang et al., 2022).

This application primarily refers to methods of interactively communicating in tasks like dialogue or human-robot collaboration. We distinguish this from creating explainable machine learning models which we address in Section 3.4.

**Applicability**   The central argument for using emergent communication to better communicate with humans comes from the fact that an emergent communication agents naturally develop competency with a wide range of linguistic phenomena. The hypothesis here is that the same functional pressures that drive the pragmatic and social aspects of human language could be replicated by sufficiently rich and embodied emergent communication environments. Thus, the emergent communication agents could not only develop the syntax and semantics of the language but also pragmatic elements in response to the environmental and social pressures. In fact, Bisk et al. (2020) argue that embodiment and interaction, beyond simply modeling static corpora, are necessary for learning to use the full depth of language. Emergent communication, then, could be a more compute-driven (and less human-feedback intensive) way of imbuing machine learning models with a full range of linguistic competency that is necessary for seamlessly interacting with humans.

**Current state**   Communicating with humans is an oft-cited potential application of emergent communication techniques, although few papers have directly experimented with it. The papers we found in the survey were proof-of-concept tasks which demonstrated some possible methods for emergent communication agents interacting with humans. One of the characteristic design choices of each paper is deciding how to structure the communication channel between the human and agents.

For the direction of human-to-agent communication, Tucker et al. (2021) map natural language to a joint embedding space with the emergent language, making the embedded natural language understandable to the agents. Li et al. (2022) have humans select embeddings directly from a labelled visualization of the embeddings (i.e., t-SNE) of emergent language messages. For the direction of agent-to-human communication, Tucker et al. (2021) visualize the embedding of agent messages in a labelled embedding space, allowing a human to determine which cluster of messages an unlabelled message belongs to (shown in Figure 6a). Mihai & Hare (2021a) show the human directly with a "message" (i.e., sketch) in a sketch-based signalling game (shown in Figure 6b). Apart from direct human-agent interaction, Tucker et al. (2022a) demonstrate machine

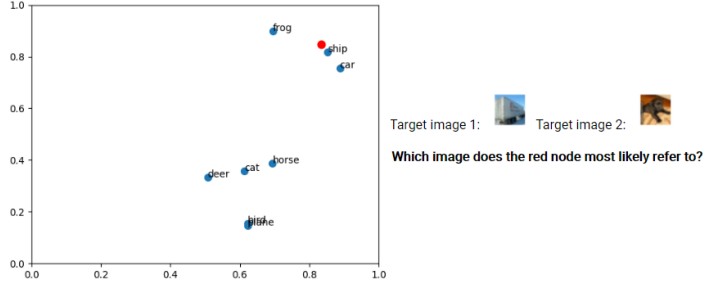
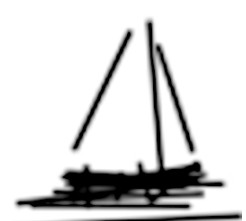

(a) User interface for interpreting the meaning of an emergent communication method from a visualization of the message's embedding (Tucker et al., 2021).

(b) Example of a "message" from a sketch-based game (from documentation of Mihai & Hare (2021a)'s code).

Figure 6: Two examples of agent-to-human emergent communication interfaces.

translation-based approach where human and emergent languages are aligned through an image captioning task.

**Next steps** The first direction for using emergent communication to augment human-computer interaction is to determine the most the natural and scalable methods and modalities for human and computers to communicate. The existing literature uses a handful of methods some of which are either unnatural and not scalable to more complex communication (e.g., interacting with concept/word embeddings). Work on non-emergent human-computer interaction can inform emergent communication research not only on methods of communication but also concerning what environments would have the potential for complex communication while still being simple enough to work with. For example, Narayan-Chen et al. (2019) present a collaborative building game in a Minecraft environment which could satisfy these criteria.

The second direction for this application is empirically demonstrating the intuitive advantages of emergent communication over more established methods for human-computer interactions. The pragmatics of interacting with humans is one of the areas with the most potential because pragmatics are inherently flexible, tied to extra-linguistic knowledge, and are more difficult to formalize than, say, syntax or semantics. Nevertheless, emergent communication could help in the more difficult regions of syntax and semantics, such as disambiguating utterances which rely on contextual knowledge or common sense reasoning.

### 3.4 Explainable machine learning models

**Description** Explainable machine learning models are those which can communicate to humans the reasons or factors behind a certain their decision. Such model are a response to deep, black-box neural models which may be able to make accurate decisions but often for opaque or seemingly arbitrary reasons. Instead it is desirable for explanations to be: (1) causally related to actual decision that is made (i.e., not a *post hoc* rationalization); (2) expressed natural language, which is one of the most effective ways to convey ideas to humans; and (3) not impose a significant negative impact on the performance of the model.

Two paradigms of explainable machine learning models illustrate solutions only satisfying some of the criteria. The first paradigm is using language generation models to generate explanations based on the hidden states of a model; while this permits the use of deep neural models, the explanations are decoupled from the actual decision since the explanation is superfluous with respect to the actual decision. The second paradigm is using explicit, interpretable steps in reasoning to the prediction (e.g., decision trees, knowledge graphs); although these explanations are now causally efficacious with respect to the prediction, it restricts the complexity of model that can be used to make the prediction. While the explanations these models generate are intrinsically related to the decisions made (e.g., the weights of a regression both explain the decision and cause it), they restrict the complexity of the model, and hence, can hamper overall performance.

*Explainable machine learning models*, in some sense, is a subclass of *Interacting with humans* (Section 3.3); here the interaction is always focused on a machine learning model communicating accurate and interpretable explanations for its decision or behavior.

**Applicability**  Emergent communication takes a radical approach to both the causal efficacy and the natural language aspect of explainable models. To illustrate this, we can describe a "deliberative ensemble of emergent communication agents". Such an ensemble would be posed a semi-adversarial game where first each member of the ensemble would generate an output for a given input. After this, the ensemble members would communicate in the emergent language to try to convince the other members of the particular output before aggregating the members' revised decisions. Given that emergent language is designed to resemble human language, the representation mismatch between natural language and the emergent language discourse is far less than natural language and the activations of a monolithic neural network. Furthermore, since the deliberation and communication among agents is critical in the final decision of the ensemble, the explanation has a direct causal link to the decision.

**Current state**  Using emergent communication for creating explainable machine learning models has only seen proof-of-concept exploration in one series of papers. Namely, Santamaria-Pang et al. (2020); Chowdhury et al. (2020a;b;c) implement and experiment with a medical image classification model which, internally, is a Lewis signalling game (Lewis, 1969). This means that the internal representations are themselves the discrete messages of an emergent language. Messages-as-internal representations, here, are intended to be a more natural modality for human working with the system than, for example, the activations of intermediate layers in the neural network.

**Next steps**  The first direction for using emergent communication for explainable machine learning models is exploring methods of generating explanations beyond the signalling game that we see in the current literature. The signalling game, while providing potentially interpretable messages, does not effectively exhibit the multi-step reasoning which (1) is most suited to the complex decisions which we would want explained, (2) is how humans generally explain themselves, and (3) is where emergent communication has the greatest potential to surpass more established methods. Such games or environments might incorporate incentives for agents to collaborate and reason sequentially using the emergent language. This reasoning process would then double as the basis for the decision and the explanation of the decision.

The second direction is incorporating state-of-the-art models into the emergent communication systems. This application, more so than others, requires that the emergent communication-based model perform comparably on downstream tasks to more established explainable machine learning models; even if the emergent communication-based models are highly explainable, they are of little practical use if they are not comparable in performance to traditional approaches. Given the size of current state-of-the-art models and inherent difficulty of training emergent communication models, this incorporation, in the near term, would likely be limited to leveraging pre-trained models which could be, at most, finetuned.

## 4   Knowledge-Driven Applications

The knowledge-driven applications of emergent communication center around the scientific fields of linguistics and cognitive science and typically concern gaining a deeper understanding of phenomena in the natural world. These applications have tend to have more remote impacts than the task-driven applications, but they also present the opportunity to gain novel insights into how humans think and use language. The primary challenges in this area come from creating emergent communication which is realistic enough to legitimately provide insight in areas where there are gaps left by more traditional techniques in linguistics and cognitive science. The first subsection below (Section 4.1) provides a summary of common themes in the "Description" and "Applicability" subsections throughout knowledge-driven applications (i.e., it is not itself an application).

### 4.1 General paradigm of knowledge-driven applications

**Description** Some of the most persistent debates in linguistics are about the degree to which language and its characteristics are the product of very specific biology (the "Chomskyan" nativist position that dominated North American linguistics in the second half of the twentieth century) or can be derived from very general mechanisms of learning (the behaviorist position that dominated North American linguistics in the first half of the twentieth century). This conflict reflects a broader debate within the social and behavioral sciences about the relative importance of "nature" (the inductive biases of the human brain) and "nuture" (operant conditioning from parents, caregivers, and other aspects of the environment) in the cognitive development of human children. Such debates are difficult to resolve because of limited access to the necessary data: the ingredients of language (nature and nuture) are largely fixed, meaning we cannot (ethically) vary them in order to determine their effects on language. This is to say, the relevant data in these debates come largely from observation and only extremely limited experimentation. The lack of true experimentation hinders the type of scientific investigation which would yield more definitive answers to these questions.

**Applicability** Emergent communication can address these unsolved problems by serving as a proxy for human language whose ingredients can be manipulated and experimented with. Emergent communication makes a suitable proxy because (1) it aims at being a faithful reconstruction of human language, and (2) this reconstruction is a reflection of its ingredients. For example, we can see the "nature vs. nurture" distinction paralleled in the distinction between the systems inside of an agent and the interaction that takes place with other agents.

Deep learning-based emergent communication is uniquely poised to serve as a proxy for human language for two reasons. First, deep learning methods are by far the closest methods to replicating human proficiency in language (as well as vision, planning, and so on). Hence, it would seem a model class of comparable power is necessary to support the emergence of a language with enough complexity to be useful for the most relevant linguistic problems. Second, deep neural networks also introduce minimal inductive bias when compared with traditional simulations and mathematical models. The behaviorist or "nurture" position can only be validated if language learning can take place without language-specific inductive biases and this is only possible in a context in which learning according to very general principles is possible, so deep learning is a natural fit for testing hypotheses about the necessity of language-specific learning mechanisms.

### 4.2 Language, cognition, and perception

**Description** This goal refers to the two-way relationship between language and cognitive (and perceptive) processes in the human brain: how language is shaped by the cognitive capacities of humans and what goes on in the brain to enable the use of language. By extension, this also includes behavior which proceeds from cognitive phenomena of interest (e.g., adjusting communication strategies based on a theory of mind). Aside from not being able to experimentally modify the brain, a major barrier in studying cognition is being able to merely *observe* the brain.

The primary way of studying language and cognition has been through laboratory experiments with humans. While we do have easy access to humans using language, the observation of the actual cognitive processes we are interested has limitations in both its direct and indirect forms. Direct observation includes using apparatus like an EEG, MEG, or fMRI; its primary disadvantages are that it requires specialized instruments, often cannot be done *in situ* and is still limited with what it can observe. Indirect observation includes methods which infer cognitive processes from external observations; for example, we might infer a limit to working memory by seeing how many digits in a long number a person can recall. The primary restrictions with indirect methods is that they, too, are very limited in what they can observe.

Some approaches to simulation for this application investigate the similarity of language models to humans in the cognitive domain (Schrimpf et al., 2020; Misra et al., 2021; Mahowald et al., 2023). These neural networks, though, are typically trained in a standard supervised or self-supervised manner (i.e., not the embodied reinforcement learning of emergent communication). Even if the model is trained with multi-modal data, the relationship between the modalities is more rigid insofar as it is restricted *a priori* by the way

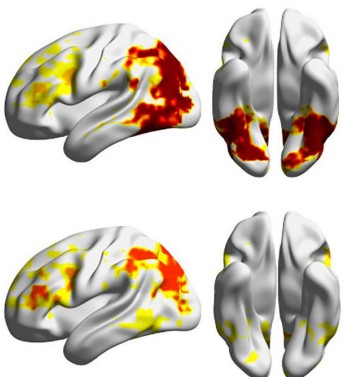

(a) Visualization of fMRI scans of the human brain in response to visual stimuli (Bracci et al., 2023).

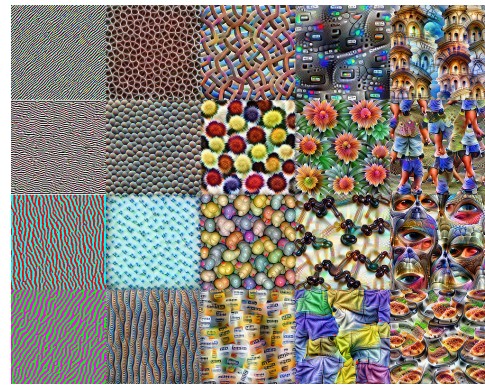

(b) Visualization of image classification CNN layers (Olah et al., 2017).

Figure 7: Measurements and visualizations of artificial neural networks are easier to make and far more flexible compared to biological networks in the human brain. This makes ANNs an attractive proxy for studying the human brain.

the model is optimized; this limits the ability to draw conclusions about human linguistic behavior where relationship between modalities is flexible and dynamic.

**Applicability**  Observing neural networks is easier that observing the results of human-subject experiments. This is because the state and processes of artificial neural networks are completely accessible, even if they are not always easy to interpret. Furthermore, any individual aspect of an artificial neural network can be manipulated, which allows for a far higher granularity in experimentation than human subjects. Compared to using language models, emergent communication agents have a more natural integration of language capabilities with other capabilities such as perception or interpersonal communication goals. This is due to the automatically learned neural-to-neural interface between between language, cognition, and perception, allowing the resulting use of language to be shaped by embodiment and pressures for useful communication.

**Current state**  The current literature in this area focuses on observing high-level principles from cognitive science and perception in the context of emergent communication systems. While these abstract facts do relate to cognitive science, they are more directly aimed at improving emergent communication techniques themselves (i.e., like an *internal* goal). Work directly applying emergent communication-trained models to particular questions within cognitive science (along the lines of Misra et al. (2021)) is largely absent.

The subtopic with the most attention in this application is the relationship between emergent communication and the agents' perception of the environment. Bouchacourt & Baroni (2018) establish a simple but important point regarding perception: neural-network based agents may successfully communicate with degenerate perceptual strategies. Namely, they show how agents which learn to play an image discrimination game with natural images are just as successful when playing with random noise images, demonstrating that we cannot simply assume that agents will learn intuitive or interpretable perceptual representations without further investigation.[10]  Nevertheless, Dessì et al. (2021) counter this pessimism by demonstrating that it is still possible for emergent communication agents to develop interpretable visual representations on their own.

Choi et al. (2018); Portelance et al. (2021) study how the balance of visual attributes in training data directly influences what attributes are actually perceived. Feng et al. (2023) look specifically at *relations* between visual elements in a referential game. More generally, Lazaridou et al. (2018); Ohmer et al. (2021b;a) study how the emergent communication is sensitive, in general, to the perception of the environment. While most papers address *visual* perception, Khazar Khorrami (2019) looks at the emergent perception of units of sound.

---

[10]This is closely related, both technically and methodologically, to adversarial inputs in computer vision research.

Deeper than perception, some work studies the agents' internal representations themselves. Sabathiel et al. (2022) look at how agents can represent numbers to themselves by interacting with their environment (e.g., an abacus). Santamaría-Pang et al. (2019) compare representations learned with supervised methods (e.g., a convolutional neural network trained on image classification) with those learned with self-supervised learning; supervised learning yields better representations, generally, but self-supervised learning can be augmented to approach the same performance. Garcia et al. (2022) discuss how a mismatch in internal representation severely reduces the effectiveness of communication.

Finally, a handful of papers have addressed cognitive strategies themselves and specifically how human-inspired inductive biases can be beneficial both for task success and for learning intuitive representations. Todo & Yamamura (2020) find that agents restricting their own learning process lead to languages with more interpretable structure; specifically, agents would discard training examples which diverged more than certain threshold from their own representations. Yuan et al. (2020); Piazza & Behzadan (2023) encourage agents to develop a theory of mind by explicitly modeling the internal states of other agents. This leads to more effective communication by introducing pragmatics into the emergent communication since agents can explicitly infer meaning from the communicative context. Masquil et al. (2022) propose adding intrinsic motivations to agents to improve communication. Finally, Cowen-Rivers & Naradowsky (2020) explore the use of world models (Ha & Schmidhuber, 2018) to improve agents' ability to handle environments with longer episodes.

**Next steps**  The next steps for this area of emergent communication are to bring the research which already explores abstract principles of cognition in emergent communication closer to the more concrete questions already present in cognitive science. This would entail using emergent communication techniques in the same vein as Misra et al. (2021) and the other papers mentioned in the "Description" section. In particular, it would be especially important to identify the differences between traditional language models and emergent communication agents in terms of their cognitive realism. This would include both ways in which language models should be limited (e.g., language models having super-human recall) as well as ways in which they need to improve (e.g., discourse coherence, factuality). Incorporating cognitive science will better illuminate where emergent communication techniques diverge from human cognition and behavior and how that might influence the resulting emergent communication.

### 4.3  Origin of language

**Description**  The origin of human language, as a task, comprises studying the environment and processes under which human language, as we recognize it today, emerged from pre-linguistic communication (e.g., methods animals use to communicate, see Figure 8). In particular, one of the biggest questions surrounding the origin of human language is whether it occurs gradually or through saltations (discussed in LaCroix (2019)). The gradualist position holds that there was no clear boundary or and no clear discontinuities between pre-linguistic communication and true human language while the saltationist position holds that, at some point, pre-linguistic communication underwent a sudden transition into human language. Addressing this particular question is major step in determining the nature of the processes explaining the origin of human language.

Since language was originally only spoken, there are no direct data which describe what happened when it evolved. Thus, any data for research come from inferential data from animal communication, and contemporary examples of language invention (e.g., creolization, Nicaraguan Sign Language). These are relatively sparse, leaving the origin of language very difficult to study. As a result, simulation is, in a way, the closest source of data to direct observation. Yet critical factors in the origin of language include complex non-linguistic elements such as perception, internal representation, and social dynamics which traditional simulations have difficulty representing.

**Applicability**  Simulation is a natural way to address processes, such as the origination of language, for which we have no (or limited) direct observations. Simulations permit not only observing these processing but counterfactually experimenting with them as well (e.g., answering "If I change variable $X$, how does $Y$ respond?"). Such experiments are necessary for scientifically distinguishing causation from mere correlation. Yet, the dependence of the origin of language on non-linguistic factors like perception, internal representations,

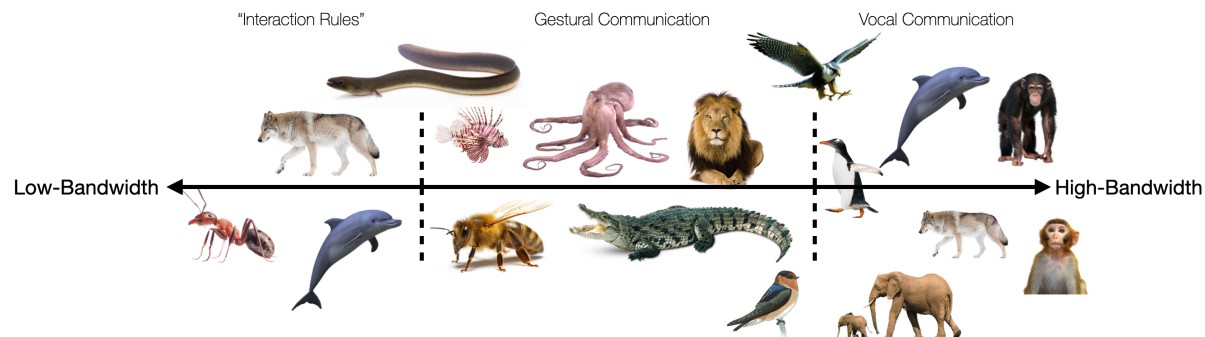

Figure 8: Illustration of the continuum of pre-linguistic communication from simple, low-bandwidth communication (e.g., ants leaving pheromone trails) to more complex, high-bandwidth systems (e.g., the vocalizations of Rhesus monkeys) (examples and figure taken from Grupen et al. (2020a)). Pre-linguistic communication systems, such as these, are an important component of studying the origin of human language.

and social dynamics indicates a significant need for simulations which integrate learning methods with a high capacity and flexibility, that is, deep neural networks. Furthermore, learning these linguistic and non-linguistic skill jointly (as opposed to, for example, using a pre-trained vision network) is also an important point of realism which emergent communication provides as it mirrors the fact that humans learn language and other cognitive skills jointly. Additionally, using neural networks allows the simulation to reflect the evolutionary pressures in the environment instead of the stipulations of a handcrafted mathematical model.

**Current state**   Work on language evolution and change comprises a few empirical papers which have used small-scale, simple environments to test specific hypotheses as well as a few position papers. The empirical papers typically use environments and tasks from prior work with the added element of transmission of language from generation to generation. For example, Grupen et al. (2021) look specifically at pre-linguistic communication (e.g., between animals) with emergent communication techniques as a foundation for the emergence of fully linguistic communication. Li & Bowling (2019); Ren et al. (2020) test the effects of *iterated learning* in emergent communication environments. Iterated learning is a framework introduced by Smith et al. (2003) as a way to reason about and explain the origin of compositionality (among other things) in human language from an evolutionary perspective on language (Kirby & Hurford, 2002; Kirby et al., 2008). The core feature of iterated learning is that when language users transmit only a subset of the language to language learners, the learners have to generalize what they have heard in order to infer the rest of the language, leading to greater systematicity and compositionality over generations.

The position papers on this topic all specifically incorporate relevant work from the linguistics side of language evolution and try to square it with the contemporary approaches of emergent communication. LaCroix (2019) compares the relative merits of gradualist and saltationist approaches to the origin of language and what bearing they have on emergent communication research, specifically arguing that the focus on compositionality might not align with gradualism. Moulin-Frier & Oudeyer (2020) highlight the opportunities and challenges of using recent advancements in multi-agent reinforcement learning for studying the origin of language. Galke et al. (2022) specifically identify the elements of current emergent communication research that must change in order to better apply to linguistically-grounded study of the origin of language.

**Next steps**   Achieving realism in emergent communication-based simulations of the origin of language must focus on closing the gap between the two data points we do actually possess: animal communication and behavior (pre-origin of language) and contemporary human language (post-origin). Thus, the pre-origin side of this entails aligning emergent communication settings with what we can currently observe in the more sophisticated varieties of animal communication, along the lines of what Grupen et al. (2021) study. Subsequently, changes to the setting would be made to elicit more sophisticated forms of communication which would ideally result in communication bearing the traits of human language (i.e., rederivation as

described in Section 2.1). Since the origin of language depends heavily on the aforementioned non-linguistic concepts, simulations will have to take into account the relevant literature in cognitive science and behavioral psychology.

Additionally, empirical implementations of the principled, interdisciplinary recommendations of the position papers (LaCroix, 2019; Moulin-Frier & Oudeyer, 2020; Galke et al., 2022) also present concrete opportunities for quickly advancing emergent communication's relevance to studying the origin of language.

### 4.4  Language change

**Description**  Languages are perpetually changing, sometimes above and sometimes below the level of conscious awareness. Language change refers to the processes which govern how language changes and develops over time in human populations. In a groundbreaking paper in language change, Weinreich, Labov, and Herzog identified five problems regarding how languages change over time Weinreich et al. (1968):

**constraints**  What constrains the transition of a language from a state $s_{t-1}$ to a successor state $s_t$? In particular, are there impossible languages that no change could produce?

**transition**  What intervening stages must exist between states $s_{t-1}$ and $s_t$? For example, do the two language varieties coexist for a time?

**embedding**  How are the observed changes embedded in the matrix of linguistic and extralinguistic concomitants of the forms in question? What other changes co-occur with the change non-accidentally?

**evaluation**  How do members of the language community subjectively evaluate the change that is underway or has occurred?

**actuation**  Why does a particular change occur at a particular point in time and space?

While human laboratory experiments have been useful in addressing some of these problems (Roberts, 2017), as have field studies and other social-scientific methodologies, emergent communication simulations provide an unprecedented means of addressing all of these problems except *evaluation*.

**Applicability**  Emergent languages in multi-agent simulations change over time. If they did not—in some respect—change, they would never develop language-like properties in the first place. Thus we can ask if they reach stable equilibria and, if so, where and why do changes occur, if at all. In answering this question, emergent communication simulations can address the *actuation problem* (one of the most difficult problems in language change). These simulations allow us to dissect the relationships between language changes and changes in the "social" and "physical" environment as well, addressing the *embedding problem*. But because emergent communication simulations give us a kind of omniscience, they also allow us to characterize the stages between stable equilibria, providing a window onto the *transition problem*. Finally, because emergent communication researchers are free to add and remove constraints on possible languages at will, such simulations allow us to address questions about whether human-like language change requires constraints on what languages are "legal" (addressing the *constraints* problem in a way that bears upon the behaviorism-nativism debate).

**Current state**  Language change has not received much attention in the literature; only two papers were found in the survey which approached the topic specifically. First, Graesser et al. (2019) study language contact, where two or more populations of agents who have developed their own language in relative isolation subsequently start communicating with each other. In particular, the experiments replicated a handful of general language contact phenomena that are known to occur with human language. First, while dialects start out as mutually unintelligible, interaction between subsets of to populations can cause convergence of all agents to a mutually intelligible language. Second, when this contact occurs, either the larger population's language will dominate and take over the smaller population's or a type of creole will form with a lower overall complexity. Finally, when there is a linear chain of populations, a continuum of mutual intelligibility

emerges where populations with fewer degrees of separation develop more similar dialects. These findings primarily address the *embedding problem* mentioned above.

Dekker & De Boer (2020) propose a set of of emergent communication experiments studying a historical instance of language change, namely morphological simplification in Alorese, a language of Eastern Indonesia. Specifically, the experiments look to determine if adult language contact can explain the loss of verb inflection in the whole language over time. The proposed approach is based on deep neural networks and proposes leveraging cognitive two cognitive mechanisms: Ullman's declarative/procedural model of language learning (Ullman, 2001b;a) and Lindblom's H&H model (Lindblom, 1990).

**Next steps**  Emergent communication studies of the transition problem have the most potential near-term progress. In particular, studies could investigate quantitatively and at scale how transition between two stable states $s_{t-1}$ and $s_t$ takes place. Specifically, one could investigate whether two languages coexist within a community of agents, with one gradually gaining currency or first dominating a subgraph of the social network, or whether changes happen abruptly across the whole population. Such studies with emergent communication could then be compared to historical examples of the transition program to verify and improve the effectiveness of emergent communication approaches.

### 4.5  Language acquisition

**Description**  Language acquisition is the process by which a human acquires the ability to use a new language. For this application, we will focus on first language acquisition because it has weightier scientific implications than second language acquisition and stands to gain more from emergent communication techniques due to how it co-occurs with the acquisition of important non-linguistic behaviors like reasoning and memory. Compared to the origin of language (Section 4.3), observational data of first language acquisition data is readily available as it always occurring in a population of humans. Compared to the cognitive and perceptual aspects of language (Section 4.2), there is more to be learned from direct observation of external behavior, making the data easier to collect. Nevertheless, data on first language acquisition is predominantly *observational*, that is, not derived from controlled, randomized experiments. Experiments which test anything more than superficial aspects of language acquisition could have drastic negative effects on human subjects and would be wholly unethical. Thus, data from more involved experimental methods on first language acquisition has to come from other sources such as neural networks trained on language data. Neural networks trained purely on *text* language data, though, fall far short of human performance given a similar amount of language data, suggesting the non-linguistic inputs might be key to replicating human language acquisition (Warstadt & Bowman, 2022).

**Applicability**  Emergent communication naturally integrates non-linguistic inputs into language (e.g., embodiment, interaction) into the acquisition of language by the neural network agents instead of stipulating ahead of time how such inputs will impact the emergent communication (Warstadt & Bowman, 2022; Bisk et al., 2020). Furthermore, the ease of observing and experimenting with neural networks vastly surpasses doing so with human subjects. These advantages of using emergent communication as a simulation technique for studying language acquisition are generally similar to those discussed in *Origin of language* (Section 4.3) and *Language, cognition, and perception* (Section 4.2) (see those sections for further details). While many of the advantages of emergent communication techniques in studying language acquisition could be derived from more traditional machine learning methods used on multi-modal data, these traditional methods are only ever *mimicking* the acquisition process that a human goes through to acquire that human language (as a first language). On the other hand, with emergent language acquisition, we can observe the selfsame acquisition process that has formed the language in the first place and not just an approximation thereof. This direct connection is important since every step in empirical reasoning which involves approximations brings with it more uncertainty in the conclusions.

**Current state**  Current literature has not often investigated language acquisition, so we will address the collected work exhaustively. At the level of individuals, current work has mainly looked at how the process of language acquisition interacts with the emergence of compositionality and other properties of language. Korbak et al. (2019; 2021) propose a developmentally-inspired curriculum which breaks down

language learning into multiple phases; they then show that this method results in more compositional emergent communication. Cope & McBurney (2022) present a method by which a new agent could acquire a pre-existing emergent language purely through observation by inferring the intentions of the observed agents. Kharitonov & Baroni (2020a) investigate a relationship in the opposite direction, looking at how the degree of compositionality of a language factors into the ease and speed of language acquisition. At a population level, Li & Bowling (2019) investigate the same relationship between ease of acquisition and compositionality in a generationally transmitted setting, arguing (in line with Smith et al. (2003)) that the pressure to acquire language from incomplete data can translate to a pressure towards compositional language. Leaving aside compositionality, Portelance et al. (2021) study the origin of shape bias, arguing that it can be explained with communicative efficiency pressures rather than inductive biases in the human or machine agents.

**Next steps**  The next steps for studying language acquisition are to demonstrate how emergent communication techniques build directly on prior work studying deep neural network-based models of language acquisition. Warstadt & Bowman (2022) mention that neural networks hold potential for studying language learning but also present a number of difficulties; thus future work in emergent communication would do well to follow existing work on the topic closely (at least for the near term). For example, Warstadt & Bowman (2020) determine that a neural network (namely BERT) is able to make structural generalizations in natural language but only after a observing more data than is developmentally realistic. Similarly, Chang & Bergen (2022) compare word acquisition in children and language models. In both cases, emergent communication could help determine if the lack of embodiment and interactivity in standard language model training explains part of why language models require significantly more data than humans to acquire the same proficiency with language.

## 4.6   Linguistic variables

**Description**  Linguistic variables are the particular phenomena in language and its use which are the subject of scientific study in linguistics. This is a catch-all application which includes all studies seeking to determine the relationships between linguistic and other linguistic/non-linguistic variables. These variables span all of the various subfields of linguistics, forming a rough low- to high-level hierarchy:

**phonology**   patterns of individual units of sound

**morphology**   patterns of individual units of meaning at the word and sub-word level

**syntax**   organization of words into meaningful structures (e.g., phrases, clauses, sentences)

**semantics**   the inherent meaning of utterances in a language

**pragmatics**   meaning derived from context cues in conjunction with semantics

**sociolinguistics**   properties of language in the context of group and social dynamics

Beyond identifying individual relationships, broader questions within linguistics concern patterns across relationships. In particular, a central question across all of the above fields, has been the degree to which linguistic variables are the product of formal properties of cognition (formalism) and to what extent they are the emergent result of language use in a communicative context (functionalism). For example, is the tendency of vowel systems to be more-or-less maximally dispersed with the formant space a result of formal universals such as a categorical phonological features that impose a straitjacket on the realization of the vowels or a result—in language evolution—of vowel distinctions that are not well-dispersed collapsing (leaving only the well-dispersed vowels behind) (Blevins, 2004).[11]

Likewise, it has been observed that prefixes and suffixes (in words that have more than one) are ordered so that those with the greatest relevance to the meaning of the root are closest to the root. This has been attributed to a formal constraint in which morphological scope mirrors syntactic scope (the Mirror Principle)

---

[11]Or, perhaps, due to a human drive to communicate as clearly as possible, given the same investment of effort (Flemming, 2013).

Baker (1985) or as a functional tendency based on a motivation, on the part of speakers, to distribute information predictably so that units of language are closest to the other units to which they are most relevant (the Relevance Principle) (Bybee, 1985). This distribution is argued to be the result of evolutionary processes emerging from attempts of language users to communicate with one another (Bybee, 1985).

The evolution of pragmatics is even less-well understood. Is contextual meaning a result of inherent principles of inference or is it an emergent property of communicative interaction? Linguists have not been able to resolve these issues experimentally because they involve simulating conversations between speakers over decades and centuries—not interactions that can be observed during an afternoon in the lab.

**Applicability**  In addition to the aforementioned applicable traits of emergent communication, there are two ways in which emergent communication is particularly applicable to studying linguistic variables. First, studying variables in any scientific discipline requires isolating these variables from confounding factors. Within emergent communication, it is possible to strip away confounding factors in ways that are often not possible when studying humans directly.

Secondly, the holistic way in which emergent communication simulates linguistic processes makes it particularly suitable to studying phenomena that span multiple levels of the linguistic hierarchy. For example, the variables relevant to the distinction between "who" and "whom" in modern English span morphology ("-m" as an affix), syntax ("who" functioning as a subject or object and "whom" as solely an object), and sociolinguistic ("whom" being perceived as formal, dated, etc.). Emergent communication, by design, allows for the interaction between many of the levels in the hierarchy without stipulating a particular way in which they interact. On the other hand, more traditional methods of modeling linguistic variables tend to be limited to just the micro or macro scale, and any interaction between these has to be determined ahead of time through handcrafted schemata, limiting the range of potential outcomes.

**Current state**  Linguistic variables, broadly construed, show up frequently in the literature as almost any property of emergent communication can be considered a "linguistic variable". For example, papers studying compositionality or grounding are addressing a relationship between *syntax* and *semantics* while papers looking at how to leverage extra-linguistic context for better communication are addressing *pragmatics*. Nevertheless, we mention papers here which directly tie into the study of human language and "linguistics" in the narrower sense. Given that emergent communication is in the stage of trying to look more like human language (cf. Section 2.1), the current literature in this application primarily focuses on recreating established linguistic phenomena in emergent communication settings. The following is list of summarizing the existing literature:

**phonology**  In contrast to most emergent communication environments which have discrete communication channels, Lan et al. (2020); Eloff et al. (2021) look at continuous channels and discretization pressures analogous to the relationship between phones and phonemes.

**syntax**  Chaabouni et al. (2019c) study whether or not emergent communication displays word-order biases akin to many human languages. van der Wal et al. (2020) analyze the output of unsupervised grammar induction applied to emergent communication.

**semantics**  Chaabouni et al. (2021); Rita et al. (2020); Luna et al. (2020) study the conditions under which Zipf's Law of Abbreviation (Zipf, 1950) is present in emergent communication. Kågebäck et al. (2018); Chaabouni et al. (2021) study the way emergent communication divides up color spaces as compared to human languages. Finally, Steinert-Threlkeld (2019) looks at the emergence of function words in emergent communication as opposed to the exclusively content-based words in most other settings.

**sociolinguistics**  Graesser et al. (2019); Kim & Oh (2021); Fulker et al. (2022) look at the formation of dialects under different conditions in networks of interacting agents. See "Current State" of Section 4.4 for Dekker & De Boer (2020).

**Next steps**  Phonology and morphology are relatively understudied in this area since most emergent communication environments assume a one-to-one correspondence between discrete symbols and "words".

The paradigm of discrete symbols-as-words precludes analyzing sub-word components since a discrete symbol has no structure. Thus, breaking away from this paradigm would open new avenues for research into the phonological and morphological aspects of emergent communication. This could be done either be simply analyzing discrete symbols as sub-word units (requiring some other definition for what constitutes a word in an emergent language), or by using a continuous communication channel with some sort of discretization pressure (such that clusters of continuous signals can be analyzed as discrete units).

Syntax and semantics are already studied in emergent communication, although this research needs to be more tightly coupled with thoroughly linguistic accounts of these phenomena instead of relying on looser, higher-level analogies with linguistics. This is a non-trivial task insofar as the definitions and models from linguistics will need to be adapted to the unique difficulties of emergent communication. For example, emergent communication can have radically different forms compared human language (or no organization at all); this means that linguistic accounts may make assumptions about the language being studied that do not necessarily hold for emergent communication (e.g., languages are, at most, mildly context sensitive). Thus, operationalizing linguistic definitions for emergent communication will require expanding their scope to account for the numerous edge cases that emergent communication presents.

For pragmatics and sociolinguistics, emergent communication environments will generally have to incorporate more agents, temporality, and embodiment. This is because these linguistic phenomena operate across many instances of language use with a common context across time and among speakers (e.g., conversational, spatial, and cultural context). In contrast, many emergent communication environments currently use single-step, simple observation, two-agent environments which preclude observing almost all pragmatic and sociolinguistic phenomena. The above point about linguistic definitions syntax and semantics requiring adaptation to emergent communication holds true for pragmatics and sociolinguistics as well since many behavior biases and heuristics we observe in humans emergent communication agents may not possess at all.

## 5 Discussion

### 5.1 Quantitative summary of results

In Figure 9, we present a quantitative summary of the categorization of papers covered in our survey. Figure 9a shows at the number of paper falling within the scope of each application, and Figure 9b further breaks the down *Linguistic variables* (Section 4.6) into the different fields of linguistics. Note that there is not a one-to-one correspondence between papers and applications, a paper may have no applications if its contributions are not properly applications or more than one application if its contribution touches on multiple areas.

### 5.2 Internal goals

The internal goals of emergent communication prove tricky for this survey since they both make up the majority of contributions in emergent communication papers but only loosely qualify as applications. Many of the papers we surveyed listed contributions along the lines of "introducing an environment where we can observe $X$ phenomenon" or "demonstrating a relationship between variable $X$ in the environment and variable $Y$ in the emergent language". The second of these was by the most common in the 245 emergent communication papers surveyed: it appeared 116 times whereas the next highest category was "related to compositionality" with only 46 papers. This is not to say that these contributions are unimportant or unnecessary, but they fail to be true applications in the sense of being a focused "goal" which a line of research can pursue. Thus, such contributions were omitted from this survey.

Aside from these non-application contributions, the topic of *metrics* was the most common. Many of these metrics, though, are not treated as applications or goals in themselves as they are introduced for the needs of the paper and do not see reuse in subsequent papers. Nevertheless, some papers do explicitly aim towards better metrics, comparing the quality of metrics in an effort to refine the tools researchers have for analyzing emergent communication (Lowe et al., 2019; Korbak et al., 2020).

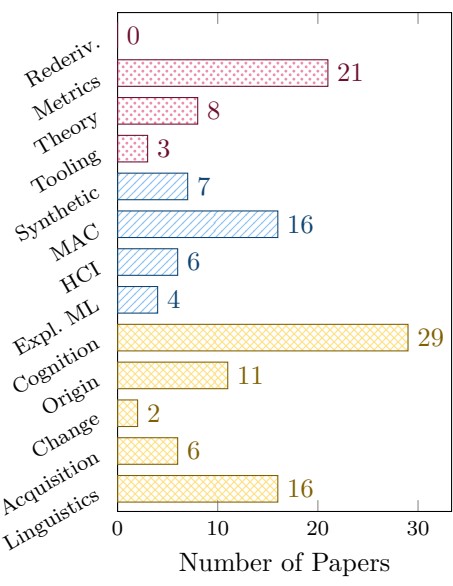

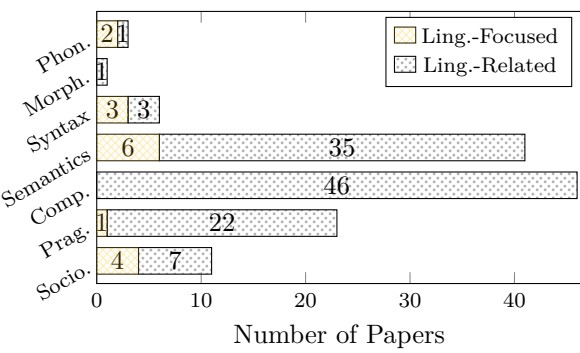

(b) Breakdown of linguistic variables papers; "Linguistic-Focused" refers to papers discussed in Section 4.6, while "Linguistic-Related" includes anything more loosely related to that area of linguistics.

(a) Number of papers corresponding to each section of this review.

Figure 9: Quantitative summary of paper topics in this survey. Abbreviations: *MAC*: multi-agent communication, *HCI*: human–computer interaction, *Expl. ML*: explainable machine learning, and *Comp.*: compositionality.

Finally, *Rederiving human language* (Section 2.1) was one goal which we included in this paper, functioning more like a position paper than a survey paper. This goal did was not explicitly pursued by any of the papers we reviewed, although it is implicit in a large number of papers, namely those which seek to align emergent communication with some human language-like quality (e.g., compositionality, Zipf's Law of Abbreviation). Nevertheless, we argue that rederiving human language should receive more attention which addresses it holistically. This is because (1) it is critical to making possible the downstream task- and knowledge-driven applications and (2) it is an effective way to interpret the other contributions falling under the "internal" umbrella.

### 5.3 Task-driven applications

Within the task-driven applications we find that multi-agent communication has received the most attention. This is generally expected as it is one the most natural applications of emergent communication, given its foundation in deep multi-agent reinforcement learning. Although some papers have addressed using emergent communication for synthetic data, it is somewhat surprising that the number is not higher since it is probably the application with the most potential for near-term success, especially in low-resource domains as a replacement for traditional synthetic data. Interacting with humans, while an important long-term goal, does not hold as much short-term promise because it is more difficult to conduct scientific studies with humans and anything short of near-human language-like emergent communication is not going to surpass other methods for interfacing with humans through natural language.

### 5.4 Knowledge-driven applications

Within the knowledge-driven the applications, we find the cognitive and core linguistic aspects of emergent communication to the most addressed. The number shown for "Cognition" in Figure 9a includes paper using a broader sense of "cognition" and "cognitive science" including topics like perception, internal representations,

and neural architectures.[12] As shown in Figure 9b, core linguistics, when given this broader interpretation is far more prevalent with over 100 in total. By comparison, language change and language acquisition of language are more niche and have fewer papers associated with them.

Within the umbrella of linguistic variables, we can see a handful of trends. First, we see generally in Figure 9b that the number of papers which address variables directly relevant to linguistics is dwarfed by the number of papers which take only a loose inspiration from linguistics. Compositionality is especially interesting in this regard as it is, by far, the most written-about topic under the broad umbrella of linguistics, yet we did not find any papers addressing it from a strongly linguistic and human language-oriented perspective.

This may be due, in part, to the fact that human languages are universally compositional and generally have similar methods of composing meaning at a broad level in comparison to many ways in which emergent communication may or may not be compositional. Aside from compositionality, semantics and pragmatics are the most studied topics. These areas of linguistics naturally line up with the most foundational aspects of emergent communication, namely figuring out what emergent languages are actually communicating (semantics) and how this meaning derives from communication strategies and environmental pressures (pragmatics). Finally, phonology and morphology have the least amount of work focused on them. One potential reason for this is that emergent communication systems are typically structured in a way to preclude phonology by using discrete communication channels and morphology by assuming discrete symbols to already be individual units of meaning (i.e., morphemes) without investigating potential subword structure further.

## 6   Conclusion

In this paper, we have given a comprehensive summary of the goals and applications of deep learning-based emergent communication research. The applications of emergent communication can roughly be categorized into those which aim at: improving emergent communication techniques themselves (internal); solving well-defined, practical problems (task-driven); and expanding human knowledge of the natural world (knowledge-driven). Each of these applications has been accompanied by a description of its scope, an explication of emergent communication's unique role in addressing it, a summary of the extant literature working towards the application, and brief recommendations for near-term research directions. Finally, we identify general trends observed in the course of surveying the applications of emergent communication.

This work has three primary goals. First, it is meant to inspire future emergent communication research by compiling the most salient areas of research into a single document with relevant work cited. Second, this work is meant to accessibly illustrate the potential applications of emergent communication to practitioners who are not as familiar with the multi-agent reinforcement learning or deep learning in general. Finally, defining the ultimate aims of emergent communication is critical to guiding the field of research itself through practices like evaluation metrics and benchmarks. Evaluation metrics require explicitly defining what a *good* or *desirable* emergent language is, and understanding what emergent communication can be used for is a foundational step. While this paper does not come close to exhausting the nuances of each of these applications, it highlights the nature and importance of applications as a whole in order to serve the future of emergent communication research.

**Acknowledgements**

We would like to thank that anonymous reviewers for their painstaking, thorough, and very constructive comments. Responding to their insightful feedback has made this article much better. This material is based on research sponsored in part by the Air Force Research Laboratory under agreement number FA8750-19-2-0200. The U.S. Government is authorized to reproduce and distribute reprints for Governmental purposes notwithstanding any copyright notation thereon. The views and conclusions contained herein are those of the authors and should not be interpreted as necessarily representing the official policies or endorsements, either expressed or implied, of the Air Force Research Laboratory or the U.S. Government.

---

[12]Although we did not perform the same focused-versus-related breakdown with cognition as was done with linguistic variables, we expect we would have found a similar divide with many cognition-related papers and only a handful of papers which focus on issues directly relevant to cognitive science (as illustrated in Figure 9b).

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

# A    Review Methods

In this section, we give a brief account of the methods used for obtaining the papers referenced in this review. As a considerable amount of the content in this review will draw from the authors' background knowledge, describing the methods does not imply that this paper is "fully reproducible". Nevertheless, presenting the process used for producing this paper can aid in understanding its context and origin.

## A.1    Collecting papers

To collect papers we searched arXiv (`https://arxiv.org/`) and Semantic Scholar (`https://www.semanticscholar.org/`) for: "emergent language", "emergent communication", "language emergence", and "communication emergence". Any paper that had a title plausibly related to emergent communication was passed along to annotation stage. Occasionally, the abstract would be skimmed at this stage, but here we aired on the side of recall and not precision.

We selected arXiv because (1) a majority of emergent communication papers are posted on arXiv and (2) the *Computer Science* archive provides a good signal to noise ratio due to the type of research that tends to be posted on arXiv. We supplemented arXiv with Semantic Scholar primarily to collect emergent communication papers that come from sources outside typical computer science discipline (as well as any CS papers which simply were not posted to arXiv). Additionally, we collected papers from all years of EmeCom[13], a series of workshops on (primarily deep learning-based) emergent communication. With very few (<5) exceptions, we gathered all of the emergent communication papers through this method. This was done primarily because it provided a good balance between overall coverage, principled methodology, and labor intensity.

**arXiv**    We searched arXiv with a disjunction of the aforementioned queries starting with the year 2015 up until present. This search was originally performed around July 1, 2022 and then again around May 4, 2023.[14] The result is approximately 4 500 entries of which 157 were selected for the next stage.

**Semantic Scholar**    The search process of Semantic Scholar was a bit more complicated because the results could not be reviewed exhaustively. This was in part because the results were sorted by relevance and also because a wider range of topics were searched. Thus, the first $100-400$ results were inspected, until further results seemed largely irrelevant to emergent communication. Similarly to arXiv, the searches were performed in two batches with the first one spanning 2015 to July 28, 2022:

- "emergent language": all fields; 100 titles reviewed: `https://www.semanticscholar.org/search?year%5B0%5D=2015&year%5B1%5D=2022&fos%5B0%5D=computer-science&fos%5B1%5D=engineering&fos%5B2%5D=linguistics&fos%5B3%5D=philosophy&fos%5B4%5D=psychology&fos%5B5%5D=sociology&fos%5B6%5D=mathematics&fos%5B7%5D=biology&fos%5B8%5D=economics&q=emergent%20language&sort=relevance`

- "emergent language": computer science; 220 titles reviewed: `https://www.semanticscholar.org/search?year[0]=2015&year[1]=2022&fos[0]=computer-science&fos[1]=engineering&fos[2]=mathematics&q=emergent%20language&sort=relevance&page=1`

---

[13]EmeCom URLs `https://sites.google.com/site/emecom2017/accepted-papers`, `https://sites.google.com/site/emecom2018/accepted-papers`, `https://sites.google.com/view/emecom2019/accepted-papers`, `https://sites.google.com/view/emecom2020/accepted-papers`, and `https://openreview.net/group?id=ICLR.cc/2022/Workshop/EmeCom#all-submissions`.

[14]Search URL for arXiv: `https://arxiv.org/search/advanced?terms-0-operator=AND&terms-0-term=emergent+language&terms-0-field=all&terms-1-operator=OR&terms-1-term=language+emergence&terms-1-field=all&terms-2-operator=OR&terms-2-term=emergent+communication&terms-2-field=all&terms-3-operator=OR&terms-3-term=communication+emergence&terms-3-field=all&classification-computer_science=y&classification-physics_archives=all&classification-include_cross_list=include&date-year=&date-filter_by=date_range&date-from_date=2015-01-01&date-to_date=2023-05-04&date-date_type=submitted_date_first&abstracts=hide&size=100&order=-announced_date_first`.

- "emergent communication": all fields; 370 titles reviewed: `https://www.semanticscholar.org/search?year[0]=2015&year[1]=2022&fos[0]=computer-science&fos[1]=engineering&fos[2]=mathematics&q=emergent%20communication&sort=relevance&page=1`

- "emergent communication": computer science: 260 titles reviewed: `https://www.semanticscholar.org/search?year%5B0%5D=2015&year%5B1%5D=2022&fos%5B0%5D=computer-science&fos%5B1%5D=engineering&fos%5B2%5D=mathematics&fos%5B3%5D=biology&fos%5B4%5D=economics&fos%5B5%5D=linguistics&fos%5B6%5D=philosophy&fos%5B7%5D=psychology&fos%5B8%5D=sociology&q=emergent%20communication&sort=relevance&page=26`

- "language emergence": computer science; 400 pages reviewed: `https://www.semanticscholar.org/search?year[0]=2015&year[1]=2022&fos[0]=computer-science&fos[1]=engineering&fos[2]=mathematics&q=language%20emergence&sort=relevance`

- "language emergence": biology, economics, linguistics, philosophy, psychology, sociology; 110 titles reviewed: `https://www.semanticscholar.org/search?year%5B0%5D=2015&year%5B1%5D=2022&fos%5B0%5D=biology&fos%5B1%5D=economics&fos%5B2%5D=linguistics&fos%5B3%5D=philosophy&fos%5B4%5D=psychology&fos%5B5%5D=sociology&q=language%20emergence&sort=relevance&page=1`

The second pass was performed on May 8, 2023, spanning 2022 and 2023:

- "emergent language": all fields; 300 titles reviewed: `https://www.semanticscholar.org/search?year%5B0%5D=2022&year%5B1%5D=2023&fos%5B0%5D=computer-science&fos%5B1%5D=engineering&fos%5B2%5D=linguistics&fos%5B3%5D=philosophy&fos%5B4%5D=psychology&fos%5B5%5D=sociology&fos%5B6%5D=mathematics&fos%5B7%5D=biology&fos%5B8%5D=economics&q=emergent%20language&sort=relevance&page=2`

- "emergent communication": computer science, engineering; 250 titles reviewed: `https://www.semanticscholar.org/search?year[0]=2022&year[1]=2023&fos[0]=computer-science&fos[1]=engineering&fos[2]=mathematics&q=emergent%20communication&sort=relevance&page=1`

- "emergent communication": computer science, eginering, linguistics, philosophy, psychology, mathematics, biology, economics; 100 titles reviewed: `https://www.semanticscholar.org/search?year%5B0%5D=2015&year%5B1%5D=2022&fos%5B0%5D=computer-science&fos%5B1%5D=engineering&fos%5B2%5D=linguistics&fos%5B3%5D=philosophy&fos%5B4%5D=psychology&fos%5B5%5D=sociology&fos%5B6%5D=mathematics&fos%5B7%5D=biology&fos%5B8%5D=economics&q=emergent%20language&sort=relevance`

- "language emergence": all fields; 300 titles reviewed: `https://www.semanticscholar.org/search?year[0]=2022&year[1]=2023&fos[0]=computer-science&fos[1]=engineering&fos[2]=linguistics&fos[3]=philosophy&fos[4]=psychology&fos[5]=sociology&fos[6]=mathematics&fos[7]=biology&fos[8]=economics&q=language%20emergence&sort=relevance`

These searches yielded 214 papers which were selected for the next stage.

## A.2  Goal categorization

Given these papers from our initial search, we reviewed the papers first to determine if they are in-scope (as described by Section 1.2) and second to categorize them according to the goals they pursued. The number of papers included and excluded is given in Table 1.

The following categories were used for the annotation of the included papers. They do not precisely line up with the section ultimately used for the paper largely because the annotation categories were determined largely *a priori* while the paper sections were determined *a posteriori*.

- Internal

  - measure properties of emergent communication
  - produce some property in emergent communication
  - other emergent communication improvement (e.g., efficiency, robustness)
  - tooling

| Category | Number of Papers |
|----------|------------------|
| *Initial Search* | 443 |
| Duplicate | 84 |
| Out-of-scope | 106 |
| Not a research paper | 5 |
| No access | 16 |
| *Included* | 232 |

Table 1: Number of papers excluded from initial search for various reasons. Duplicate papers were either overlaps between different sources or papers that were substantially similar and by the same authors.

- – theoretical frameworks
- Task-driven
  - – artificial general intelligence, better NLP
  - – replication of natural language
  - – alternative data source/paradigm
  - – robust multiagent communication
  - – explainable models
  - – synthetic data for evaluation
  - – communicating with humans
- Knowledge-driven
  - – increase understanding of language in general
  - – evolution of language:
  - – fundamentals of language (e.g., phonology, lexicon, syntax)
    - ∗ phonology
    - ∗ syntax
    - ∗ semantics
    - ∗ compositionality
    - ∗ morphology
    - ∗ pragmatics
    - ∗ sociolinguistics
  - – language acquisition
  - – cognitive science and language
    - ∗ perception

Some of these categories were eventually discarded since they either did not receive much attention in the literature or the category itself was too vague to productively discussed. A quantitative summary of the categorization after remapping them to section in the paper is for each paper is presented in Section 5.1.

For the majority of papers, we would read the abstract, introduction, and conclusion in order to assign the proper categories; this would take, on average, 6 minutes to complete per paper. These sections are the most common places for describing the broader applications and contributions of the papers. Paper were reviewed more thoroughly as needed to determine the proper categories. Determining which papers to highlight in the body of this paper depended on the application. For applications with a small number of papers, we were able to exhaustively discuss the applicable papers. For applications with many papers, we highlighted a representative sample of the papers which best illustrated that application.

## B Complete List of Reviewed Papers

**Rederiving human language** *No papers.*

**Metrics for emergent communication**  Bogin et al. (2018); Bosc & Vincent (2022); Bosc (2022); Chaabouni et al. (2020; 2022); Denamganaï et al. (2023); Guo et al. (2021; 2020); Korbak et al. (2020; 2021); Kuciński et al. (2020a); Lowe et al. (2019); Mu & Goodman (2021); Perkins (2021a); Resnick et al. (2020); Thomas & Saad (2022); Tucker et al. (2022b); Verma & Dhar (2019); van der Wal et al. (2020); Yao et al. (2022)

**Theoretical models**  Boldt & Mortensen (2022b;a); Eecke et al. (2023); Khomtchouk & Sudhakaran (2018); Ren et al. (2020); Resnick et al. (2020); Rita et al. (2022b); Tucker et al. (2022b)

**Tooling**  Denamganaï & Walker (2020b); Ikram et al. (2021); Kharitonov et al. (2019); Perkins (2021b)

**Synthetic language data**  Dessì et al. (2021); Downey et al. (2022); Li et al. (2020); Mu et al. (2023); Santamaría-Pang et al. (2019); Steinert-Threlkeld et al. (2022); Yao et al. (2022)

**Multi-agent communication**  Bullard et al. (2021; 2020); Chen et al. (2022); Cope & Schoots (2021); Karten et al. (2023); Li et al. (2022); Mahaut et al. (2023); Masquil et al. (2022); Mul et al. (2019); Piazza & Behzadan (2023); Thomas & Saad (2022); Tucker et al. (2021); Wang et al. (2022); Xiang et al. (2023)

**Interacting with humans**  Hagiwara et al. (2021); Karten et al. (2022c); Li et al. (2022); Mihai & Hare (2021a); Tucker et al. (2021; 2022a)

**Explainable machine learning models**  Chowdhury et al. (2020b;c;a); Santamaria-Pang et al. (2020)

**Language, cognition, and perception**  Bouchacourt & Baroni (2018); Chaabouni et al. (2021); Choi et al. (2018); Cowen-Rivers & Naradowsky (2020); Dekker & De Boer (2020); Denamganaï et al. (2023); Dessì et al. (2021); Feng et al. (2023); Grupen et al. (2020b); Hagiwara et al. (2021); Herrmann & VanDrunen (2022); Kågebäck et al. (2018); Lazaridou et al. (2018); Mahaut et al. (2023); Ohmer et al. (2021b); Masquil et al. (2022); Mihai & Hare (2021b); Ohmer et al. (2021a); Ossenkopf et al. (2022); Patel et al. (2021); Piazza & Behzadan (2023); Portelance et al. (2021); Sabathiel et al. (2022); Todo & Yamamura (2020); Yuan et al. (2021; 2020); Zubek et al. (2023)

**Origin of language**  Chaabouni et al. (2020); Cogswell et al. (2019); Dagan et al. (2020); Dekker & De Boer (2020); Galke et al. (2022); Grossi & Ross (2017); Grupen et al. (2021); LaCroix (2019); Moulin-Frier & Oudeyer (2020); Ohmer et al. (2021a); Ren et al. (2020)

**Language change**  Dekker & De Boer (2020); Graesser et al. (2019)

**Language acquisition**  Cope & McBurney (2022); Kharitonov & Baroni (2020a); Korbak et al. (2019; 2021); Li & Bowling (2019); Portelance et al. (2021)

**Linguistic variables, phonology (focused)**  Eloff et al. (2021); Lan et al. (2020)

**Linguistic variables, phonology (related)**  Eloff et al. (2021); Lan et al. (2020); Verma & Dhar (2019)

**Linguistic variables, morphology (focused)**  *No papers.*

**Linguistic variables, morphology (related)**  Mihai & Hare (2021a)

**Linguistic variables, syntax (focused)**  Chaabouni et al. (2019c); Ueda et al. (2022); van der Wal et al. (2020)

**Linguistic variables, syntax (related)**  Bosc & Vincent (2022); Chaabouni et al. (2019c); Resnick et al. (2020); Słowik et al. (2020b); Ueda et al. (2022); van der Wal et al. (2020)

**Linguistic variables, semantics (focused)**   Chaabouni et al. (2019a; 2021); Kågebäck et al. (2018); Luna et al. (2020); Rita et al. (2020); Steinert-Threlkeld (2019)

**Linguistic variables, semantics (related)**   Bosc & Vincent (2022); Bouchacourt & Baroni (2019); Chaabouni et al. (2020; 2019a; 2021); Cope & Schoots (2021); Dessì et al. (2019); Garcia et al. (2022); Grupen et al. (2020b); Guo et al. (2021; 2020); Guo (2019b); Guo et al. (2019); Herrmann & VanDrunen (2022); Kågebäck et al. (2018); Kharitonov & Baroni (2020a); Kharitonov et al. (2020a); Khomtchouk & Sudhakaran (2018); Korbak et al. (2019); Kågebäck (2018); Lazaridou et al. (2016); Lin et al. (2021); Luna et al. (2020); Ohmer et al. (2021b); Mihai & Hare (2021b; 2019); Mu & Goodman (2021); Ohmer et al. (2021a); Portelance et al. (2021); Qiu et al. (2021); Rita et al. (2020); Sabathiel et al. (2022); Steinert-Threlkeld (2019); Słowik et al. (2020a); Tucker et al. (2021; 2022b;b); Unger & Bruni (2020); Yu et al. (2022); Zhang et al. (2019); Zubek et al. (2023)

**Linguistic variables, compositionality (focused)**   *No papers.*

**Linguistic variables, compositionality (related)**   Auersperger & Pecina (2022); Bogin et al. (2018); Bosc & Vincent (2022); Bosc (2022); Chaabouni et al. (2020; 2022); Chen et al. (2023); Choi et al. (2018); Cogswell et al. (2019); Denamganaï & Walker (2020a); Denamganaï et al. (2023); Galke et al. (2022); Garcia et al. (2022); Guo et al. (2020); Guo (2019b); Guo et al. (2019); Havrylov & Titov (2017); Hazra et al. (2021; 2020); Karten & Sycara (2022); Keresztury & Bruni (2020); Kharitonov & Baroni (2020a); Korbak et al. (2021); Kottur et al. (2017); Kuciński et al. (2021; 2020a); LaCroix (2019); Lan et al. (2020); Lazaridou et al. (2018); Li & Bowling (2019); Liang et al. (2020); Luna et al. (2020); Mordatch & Abbeel (2018); Ohmer et al. (2022; 2021a); Perkins (2021a); Ren et al. (2020); Resnick et al. (2020); Rita et al. (2022a;b); Steinert-Threlkeld (2020); Słowik et al. (2020b); Thomas & Saad (2022); Ueda et al. (2022); Xu et al. (2022); Todo & Yamamura (2020)

**Linguistic variables, pragmatics (focused)**   Kang et al. (2020)

**Linguistic variables, pragmatics (related)**   Blumenkamp & Prorok (2020); Bouchacourt & Baroni (2019); Bullard et al. (2021; 2020); Cao et al. (2018); Eccles et al. (2019); Evtimova et al. (2017); Kalinowska et al. (2022b;a); Kang et al. (2020); Karten et al. (2023); Kolb et al. (2019); Leni et al. (2018); Lipinski et al. (2022); Lowe et al. (2019); Masquil et al. (2022); Mordatch & Abbeel (2018); Noukhovitch et al. (2021); Ossenkopf et al. (2022); Ossenkopf (2019); Piazza & Behzadan (2023); Yu et al. (2022); Yuan et al. (2020)

**Linguistic variables, sociolinguistics (focused)**   Dekker & De Boer (2020); Fulker et al. (2022); Graesser et al. (2019); Kim & Oh (2021)

**Linguistic variables, sociolinguistics (related)**   Dekker & De Boer (2020); Fitzgerald (2019); Fulker et al. (2022); Galke et al. (2022); Graesser et al. (2019); Grupen et al. (2022); Gupta & Dukkipati (2019); Kim & Oh (2021); Li & Bowling (2019); Liang et al. (2020); Rita et al. (2022a)

**No applications**   Boldt & Mortensen (2022c); Brandizzi et al. (2021); Carmeli et al. (2022); Foguelman et al. (2021); Gaya (2017); Guo (2019a); Gupta et al. (2020); Hagiwara et al. (2019); Kajić et al. (2020); Karch et al. (2022); Karten et al. (2022a;b); Lannelongue et al. (2019); Lazaridou & Baroni (2020); Lee et al. (2018); Lipowska & Lipowski (2018); Yat Long Lo (2022); Lo & Sengupta (2022); Lowe et al. (2020); Nevens et al. (2020); Brandizzi & Iocchi (2022); Raviv et al. (2022); Sirota et al. (2019); Taniguchi et al. (2022); Wang et al. (2021); Wieczorek et al. (2023)

