# OpenReview forum: "A Review of the Applications of Deep Learning-Based Emergent Communication"
_TMLR — Accepted by TMLR_

### Review · Reviewer_6hRL · 2023-09-24

**Summary Of Contributions:**

This paper provides a survey of existing work in emergent language within the ML community, and a discussion of the applications of this work and future directions.

**Audience:**

Yes

**Broader Impact Concerns:**

I would like somewhat more discussion on broader impacts wrt. generalization of EL to the significant variation in human language use across and within language communities. There is a bit of discussion on low-resource languages, but a clear discussion of how existing work engages with existing human languages (e.g., what languages do they evaluate against?) and variance across _individual_ language agents within a community is not discussed at all.

**Claims And Evidence:**

Yes

**Requested Changes:**

There are some assumptions that I think this survey is making which could be stated explicitly, and preferably discussed in depth. Perhaps the paper is not making these claims, and pushback on them would be very welcome:
* That it is possible to find and model all of the necessary conditions of language emergence for proper modeling (Section 2.1)
* That we know exactly what fundamental features of "language" (and even the scope of the term "language") are. These features are contested within the field of linguistics; in light of this, how do we maintain a flexible definition of "rederiving language"?
* That standardized tooling will be an ideal place to start for EL research. My concern here is that once experimental settings / environments become "standard", they start to encode subtle assumptions / constraints into all the work that builds on top of them, and thus also embeds these assumptions / constraints into findings on EL and proposed models.
* That EL could be useful for studying low-resource languages. I find it hard to believe that any EL protocol, even if it is _generally_ successful at synthesizing a humanlike language, would be able to reconstruct a _particular_ low-resource language. This concern also applies to modeling the long tail of language: how could we be sure that what we are synthesizing in this long tail _actually_ aligns with a particular language's long tail, and is thus useful for studying real human languages?
* That scaling up language data by generating more EL data in a particular environment will suffice. As we scale up, the Zipfian distribution must be preserved, and this means scaling the long tail via innovating new terms / uses of language. It's unclear to me that with current EL tooling this would be possible, especially in limited / synthetic environments. My intuition is that the environment itself must scale in terms of complexity.
* That training models in EL setups will recover the same kinds of pragmatic skills that would be necessary in real human-agent interaction. (Section 3.3)
* That we should care about answering the question of whether cognition is derived innately or via experience (nature vs. nurture). What motivates this debate? (I am not saying that it's not an important question, but I feel it's important to describe here _why_ we should care about the answer. There certainly exist troubling motivations for answering this question; e.g., attempts to prove certain skills are innate thus justifying discrimination)

Some of the claims are ambiguous or unsubstantiated:
* "Resemblance to human language is one of the defining characteristics of emergent language" --> is this a defining characteristic of _all emergent languages_, or is it something that we _want_ to be a defining characteristic?
* "There are a significant number of papers that look at individual components..." Does "look at" mean analyze or explicitly pursue rederivation of? Including citations here would be useful.
* "understanding facets of language in isolation (i.e., divide and conquer) can yield easier research questions" -- can an example be given here?
* It wasn't clear to me what the contribution of the analogy in statistics in Section 2.2 adds to discussions in EL; an example within EL would help here.
* EL is "more interpretable" than other communication protocols (Section 3.2).
* There is a bit of a circular argument going on that (a) EL should be pursued due to its applicability to other domains / questions (e.g., Sections 3 and 4), and a key next step to this is to (b) show that EL has applicability in a task that shows the importance of EL (in "next steps" of Section 3.2). Perhaps this is true just for Section 3.2., and other applications have clear evidence of EL support, but this argument seems odd to me.
* "The same function pressures that drive the pragmatic and social aspects of human language _can be_ replicated by sufficiently rich and embodied emergent language environments" --> theoretically yes, but is work that is not actually doing this just going to end up misleading?
* "first language acquisition... stands to gain more [than second language acquisition] from emergent language techniques" --> why?

Minor clarifying (rhetorical) questions:
* In 1.1, the system contains multiple agents -- could one of these agents be human? I didn't see any work highlighted that studies emergence / change of language in human-agent language-based interaction (e.g., Kojima et al. 2021, Suhr and Artzi 2022, Wang et al. 2017)
* In 1.1, there is a distinction between the protocol and the properties of a channel. How are these distinguished? I could imagine that constraints on a channel (e.g., that discrete tokens must be used) constrain the protocol such that these are difficult to disentangle. Perhaps the protocol is some particular instantiation within a manifold of protocols that could be supported by a channel, whose differences along the manifold are arbitrary (e.g., what particular form grounds to a particular referent)
* In 1.1: Are there exceptions to EL work using deep learning methods? E.g., in 1.2., Werner and Dyer 1991 propose a "connectionist artificial neural network" -- it seems the distinction with deep learning here is relatively subtle (i.e. that it must be learned via gradient descent). Is there a significant body of work like this that is exceptional to the definition of "deep learning"?
* How does the development of theoretical models of language emergence intersect with the challenges of complex systems modeling?
* How is AlphaGo analogous to generating synthetic language data?
* In 3.1 Current state, a lot of relevant work needs more elaboration. E.g., "relevant two papers in our survey" which papers are these? What does it mean to have baselines that are "stochastic well-balanced brackets and Spanish"? What does it mean to "capture more information"? And what is happening with video captions? It's hard to get anything from these references to current work without actually reading the papers cited to even know what task they are studying.
* In 3.1 Next steps: "different relative merits" what are these merits?
* 3.3 Current state also highlights existing work but it's hard to determine exactly what these papers are doing as the setups are not described in detail.
* 3.4 Next steps: it's unclear to me how the second paragraph here is related to explainability
* 4.1 -- why is Current state and Next steps missing?
* What is "true multi-modal integration" (Section 4.2) and what would it look like to have this true integration?
* Section 4.3 touches on a sort of bootstrapping problem in the origins of language: that representations of non-linguistic perception / action and representations of language are deeply intertwined. More discussion on this would be interesting.
* How could we know if our findings in EL actually generalize to human language change? (Section 4.4)
* Why does section 4.5 refer to non-linguistic perception/action/etc. as "pre-linguistic inputs" rather than "non-linguistic inputs"?
* "advantages of using [EL] as a simulation technique for studying language acquisition are generally similar to those discussed..." in what way? Would like more elaboration on this instantiated in language acquisition research.

Minor stylistic comments:
* It is a bit jarring for the first and last sentence in the intro's first paragraph to be directly copied from the abstract.
* The confusion around "intermediate _applications_" could be alleviated by just using a different term, e.g., distinguishing "internal" or "inward-facing" advancements from "external" applications.
* "evaluation" metric as distinguished from simple "metric" is confusing to me. Perhaps "success", "holistic", "overall" metric?

Typos:
* "wholistic" --> "holistic"
* In 2.1 Applicability: "ubiquitous part research" --> "ubiquitous part of research"
* "distinct situation (e.g., Starcraft..." --> "distinct situations"
* "full depth language" --> "full depth of language"
* "begin able" --> "being able"
* "as an task" --> "as a task"
* "look specifically pre-linguistic communication" --> "look specifically at pre-linguistic communication"

Paper citations from above:
* Kojima et al. 2021 (TACL): Continual learning for grounded instruction generation by observing human following behavior
* Suhr and Artzi 2022: Continual learning for instruction following from realtime feedback
* Wang et al. 2017 (ACL): Naturalizing a programming language via interactive learning

**Strengths And Weaknesses:**

Strengths:
* This is a very comprehensive overview of emergent language. I found it to be a useful resource myself for finding papers that interest me, although my area is not directly in emergent language.
* The paper is well-written (with some minor typos) and easy to understand.
* I liked the connection to complex systems sciences and would have loved more elaboration / discussion on this, including the challenges that we may face if we do see EL as a complex systems science (e.g., Holtzman et al. 2023 elaborates this analogy for generative language models).

Weaknesses:
* It's unclear whether a survey like this one is appropriate for publication in TMLR, as there are no empirical or theoretical results.
* The paper makes many unsubstantiated or ambiguous claims, and sometimes uses analogies, that could use more elaboration or citations. Please see "requested changes" for details.
* Some of the terminology used could also use elaboration.
* I would like more elaboration on how choices in environment design (e.g. in (standardized) tooling) might influence the findings on EL. Some of this discussion comes through in Section 3.2 Applicability, but I think it's appropriate to put in the Tooling section (with more elaboration).
* I noticed that a lot of papers in the bibliography are not associated with a publication venue. If they are preprints, this should be clear in the bib entry.

Papers:
* Holtzman et al. 2023: Generative models as a complex systems science

---

> ### Author Response · Authors · 2023-10-14
> **Reply (1/4)**
>
> Thank you for the very detailed and helpful review.  We believe the review
> is in tune with overall intent of the paper and that the critique and
> recommendations are appropriate and helpful.  Thus, if anything is left out in
> this reply, we intend to address the concern in our edits.
>
> We will __bold__ any parts of the reply where we are asking a question or need
> clarification.
>
>
> > [whether a survey is appropriate for TMLR]
>
> Although TMLR papers tend to be focused on theoretical and/or empirical
> results, we believe the paper fits both the acceptance criteria of being
> interesting to TMLR's audience as well a supported with high-quality evidence.
>
> > [preprint citations are unclear]
>
> We will fix arXiv's BibTeX exports to provide more informative entries in the
> manuscript.
>
> ## Re: survey's assumption being made explicit
>
> We will revise the paper in accordance with the following concerns and
> responses.
>
> > That it is possible to find and model all of the necessary conditions of
> > language emergence for proper modeling (Section 2.1).
>
> This is an open question which incorporates aspects of "rederiving language"
> (Sec 2.1) and the "origin of language" (Sec 4.3).  We see EL as one of the
> primary tools for _determining_ what the necessary conditions are.  However, if
> it is not possible for ML methods to model the necessary conditions of language
> emergence in the first place, this would represent a negative result for one of the strongest predictions of emergent language (i.e., it is possible to rederive human language from functional advantage).
>
>
> > That we know exactly what fundamental features of "language"...
>
> We do not assume that the definition of language is settled in any definitive
> way, but we do believe that a rougher definition is possible that distinguishes
> seemingly random emergent communication from EC that is meaningfully similar to
> human language. Certainly, though, linguistic typologists have studied the
> whole range of actually-occuring human languages and characterized much of what
> they have in common (as well as the manifold ways in which they can differ), so
> there is a foundation from which claims about the fundamental features of
> (human) language can proceed. We will provide further references to the
> linguistic literature addressing this question.
>
>
> > That standardized tooling will be an ideal place to start for EL research. My
> > concern here is that once experimental settings / environments become
> > "standard", they start to encode subtle assumptions / constraints into all
> > the work that builds on top of them, and thus also embeds these assumptions
> > / constraints into findings on EL and proposed models.
>
> We do not intend to make this assumption, although you are correct that we do
> not illustrate the downsides/limitations/concerns of this application.  We rather
> intend to say that "this is _a_ goal" and not necessarily that this is the best
> goal to pursue.  We will incorporate the aforementioned concerns to give a more
> balanced presentation of this goal.
>
>
> > That EL could be useful for studying low-resource languages. I find it hard
> > to believe that any EL protocol, even if it is generally successful at
> > synthesizing a humanlike language, would be able to reconstruct a particular
> > low-resource language.
>
> In the particular case of a low-resource language, the assumption is that there
> would be some degree of control over the formation of the EL which would permit
> us to bias its structure given the limited resources we do have for a LR
> language (e.g., a lexicon, a grammar). This is not to say that the EL would be
> a copy of the low-resource language; rather, it would have parallel structural
> properties.  Thus, EL is meant to be a method of data augmentation rather than
> approximating the LR language from nothing.
>
>
> > ... This [above] concern also applies to modeling the long tail of language:
> > how could we be sure that what we are synthesizing in this long tail actually
> > aligns with a particular language's long tail, and is thus useful for
> > studying real human languages?
>
> For high resource languages, this would be possible to verify using data-driven
> methods of comparison.  For example, we could compare emergent langugae to
> human-language for transfer learning to handcrafted synthetic data for transfer
> learning (e.g., a PCFG).  Synthetic data can capture well the most common core
> patterns in language but fails to capture the large number of infrequent
> phenomena.  If EL is able to outperform synthetic data, it could indicate
> (speaking abductively) that emergent language has done this, in part, by
> capturing the long tail.

---

> ### Author Response · Authors · 2023-10-14
> **Reply (2/4)**
>
> > That scaling up language data by generating more EL data in a particular
> > environment will suffice. As we scale up, the Zipfian distribution must be
> > preserved, and this means scaling the long tail via innovating new terms
> > / uses of language. It's unclear to me that with current EL tooling this
> > would be possible, especially in limited / synthetic environments. My
> > intuition is that the environment itself must scale in terms of complexity.
>
> We do not intend to make this assumption, and we agree that complexity must
> increase with "scale" as well.  __What part of the paper does this concern
> apply to in particular?__
>
>
> > That training models in EL setups will recover the same kinds of pragmatic
> > skills that would be necessary in real human-agent interaction. (Section 3.3)
>
> Yes, the assumption is that EL setups could and must do this _to be practical in
> this regard_, not necessarily that they do this now.
>
>
> > That we should care about answering the question of whether cognition is
> > derived innately or via experience (nature vs. nurture). What motivates this
> > debate? (I am not saying that it's not an important question, but I feel it's
> > important to describe here why we should care about the answer. There
> > certainly exist troubling motivations for answering this question; e.g.,
> > attempts to prove certain skills are innate thus justifying discrimination)
>
> The question of whether cognition emerges from a tabula rasa experiencing the
> world, or is primarily the product of articulated and in-build neurological
> endowments is important precisely because the stakes are so high. You mention
> one example: the no-longer-dormant debate about the genetic basis for cognitive
> capcities. The degree to which language behavior, and other cognitive skills,
> emerge from interaction rather than a priori stipulation is the degree to which
> we can hope that by changing variations in the social environment, we can
> foster a richer cognitive repertoire. If the strongest predictions of the
> emergent language hypothesis hold true, it would be striking evidence that it
> is not necessary to invoke (genetically) a priori biases to account for the
> sophistication of human language behavior (and by extension, large swaths of
> human cognition). We did not elaborate on this subject in the paper, __but if
> you believe that it is important to incorporate
> such a discussion, we could add the relevant details.__
>
> ## Re: Some of the claims are ambiguous or unsubstantiated
>
> > "Resemblance to human language is one of the defining characteristics of
> > emergent language" --> is this a defining characteristic of all emergent
> > languages, or is it something that we want to be a defining characteristic?
>
> It defines them as a goal/desirable characteristic, not necessarily as an actuality.
>
> > "There are a significant number of papers that look at individual
> > components..." Does "look at" mean analyze or explicitly pursue rederivation
> > of? Including citations here would be useful.
>
> Explicitly pursue rederivation of, but only in a limited scope, not treating
> it as integrated into human language.  Citations will be added.
>
> > "understanding facets of language in isolation (i.e., divide and conquer) can
> > yield easier research questions" -- can an example be given here?
>
> Yes.  For example, the broad question of "what pressures make language
> compositional" could be broken down into:
> - Perceptual pressures: "Emergence of Linguistic Communication from Referential
>   Games with Symbolic and Pixel Input" (Lazaridou et al., 2018)
> - Modelling ("cognitive") constraints "Capacity, Bandwidth, and
>   Compositionality in Emergent Language Learning" (Resnick et al., 2019),
>   "Natural Language Does Not Emerge ‘Naturally’ in Multi-Agent Dialog" (Kottur
>   et al., 2017)
> - Learning dynamics: "Compositional Languages Emerge in a Neural Iterated
>   Learning Model" (Ren et al., 2020)
> - Information theoretic: "Entropy Minimization In Emergent Languages"
>   (Kharitonov et al., 2019)
>
>
> > It wasn't clear to me what the contribution of the analogy in statistics in
> > Section 2.2 adds to discussions in EL; an example within EL would help here.
>
> We can just use "compositionality" here instead.  Compositionality most often
> refers to the general notion that "the meaning of a composite message is
> a function of the meanings of individual parts", but this notion is
> underspecified.  For example, it does not specify if meaning rests in the
> interpretation of the speaker, listener, or both; it does not specify what
> limits might exist on functions used to combine meaning; etc.
>
>
> > EL is "more interpretable" than other communication protocols (Section 3.2).
>
> __Please clarify your concern.  Is it that we state EL is more
> interpretable without further justification?  Or that it is not clear what we
> mean by "more interpretable" in the first place?__

---

> > ### Comment · Reviewer_6hRL · 2023-10-24
> > **Clarifications**
> >
> > Hi, in this comment I'll respond to the clarification questions bolded in this part of the response. Thank you for your response!
> >
> > > We do not intend to make this assumption, and we agree that complexity must increase with "scale" as well. What part of the paper does this concern apply to in particular?
> >
> > The point seemed to come up in 3.1, but I can't find the exact wording that inspired this comment. Perhaps I was putting a couple pieces together: (a) synthetic data is good because we can generate a lot of it, and (b) we can generate synthetic data with EL -> using a single EL setting and scaling up training data by sampling more and more synthetic data from it is a solution to the data scarcity problem. But it seems you didn't intend this inference to be made.
> >
> > > if you believe that it is important to incorporate such a discussion, we could add the relevant details
> >
> > I don't think it is critical to include, but my comment was just a response to the appeal to answering this question as a motivation for EL. Totally agree that if studies with EL further suggest environmental / interactional factors are critical to language development, this disproves nasty arguments for fixed cognitive capacities. I am just wondering with my question whether engaging with this debate at all is a good idea (except, of course, to disprove these arguments and biased policies built upon them). Anyway, this is not very relevant to the paper, so feel free to ignore :)
> >
> > > Please clarify your concern. Is it that we state EL is more interpretable without further justification? Or that it is not clear what we mean by "more interpretable" in the first place?
> >
> > I think both of these. Interpretability is underdefined (in general in the field, but also in this paper), and thus it's unclear why EL is considered more interpretable -- a more concrete example would help here.

---

> ### Author Response · Authors · 2023-10-14
> **Reply (3/4)**
>
> > There is a bit of a circular argument going on that (a) EL should be pursued
> > due to its applicability to other domains / questions (e.g., Sections 3 and
> > 4), and a key next step to this is to (b) show that EL has applicability in
> > a task that shows the importance of EL (in "next steps" of Section 3.2).
> > Perhaps this is true just for Section 3.2., and other applications have clear
> > evidence of EL support, but this argument seems odd to me.
>
> Regarding (a), we intend to convey that EL should be pursued because of its
> _potential and intuitive_ applicability to downstream tasks.  In the case of
> next steps for multi-agent communication (Sec 3.2), we are recommending that
> finding a niche problem that highlights the applicability of EL.  This distinct
> from many of the other "Next steps" which instead focus on the particular
> shortcomings of EL which preclude finding a niche application in the first
> place.  In short, "(a) EL should be pursued due to its promising potential
> applicability, and (b) in the case of multi-agent communication, a productive
> near-term research program is finding a niche problem".  __Let us know if this is
> missing the concern here.__
>
>
> > "The same function pressures that drive the pragmatic and social aspects of
> > human language can be replicated by sufficiently rich and embodied emergent
> > language environments" --> theoretically yes, but is work that is not
> > actually doing this just going to end up misleading?
>
> __By this, are you asking whether or not work in simplified environments is
> ultimately informative to richer environments?__  This is an important and open
> question in EL research amd only further research will answer it.
>
>
> > "first language acquisition... stands to gain more [than second language
> > acquisition] from emergent language techniques" --> why?
>
> Namely, first language acquisition:
> - Represents a greater shift in behavior: using no language to using language
>   vs. mono- to bilingual; the acquisition of language may be necessary for
>   supporting other higher cognitive functions which are characteristic of
>   humans.
> - It is more complex in how it co-occurs with the acquisition of broader skills
>   (e.g., pragmatic inference, social skills) in a much more complex environment
>   (e.g., from parents vs. in a classroom, on the job, or on Duolingo)
>
>
> ## Minor clarifying (rhetorical) questions:
>
> > In 1.1, the system contains multiple agents -- could one of these agents be
> > human? ...
>
> For the purposes of this paper, no. We cover only ELs which are generated
> solely by populations of computer agents. However, some past research has
> examined emergent language settings in which one agent was a human.
>
>
> > In 1.1, there is a distinction between the protocol and the properties of
> > a channel. How are these distinguished? I could imagine that constraints on
> > a channel (e.g., that discrete tokens must be used) constrain the protocol
> > such that these are difficult to disentangle. Perhaps the protocol is some
> > particular instantiation within a manifold of protocols that could be
> > supported by a channel, whose differences along the manifold are arbitrary
> > (e.g., what particular form grounds to a particular referent)
>
> We understand what you are saying at a basic level, but __could you point out where this
> distinction needs to be made?__  Is it primarily that there is not a definitive
> line one can draw between constraining the communication channel and the
> protocol that eventually emerges on that channel? We agree that constraints on
> the channel affect the protocols that will develop and that, thus, there is
> a potential causal dependency of protocol upon channel constraints, but these
> are nevertheless distinct entities.
>
>
> > In 1.1: Are there exceptions to EL work using deep learning methods? E.g., in
> > 1.2., Werner and Dyer 1991 propose a "connectionist artificial neural
> > network" -- it seems the distinction with deep learning here is relatively
> > subtle (i.e. that it must be learned via gradient descent). Is there
> > a significant body of work like this that is exceptional to the definition of
> > "deep learning"?
>
> We are not as familiar with the pre-deep learning antecedents of DL-based EL as
> with current DL-based methods, but we would say that there does not need to be
> a clear line between EL with DL and EL through other methods.  We draw a hard
> line here primarily for practical purposes of the survey (i.e., we had to draw
> the line somewhere).
>
>
> > How does the development of theoretical models of language emergence
> > intersect with the challenges of complex systems modeling?
>
> This is a topic we will need to look into more and address accordingly in the
> paper. __Do you have any specific literature in mind?__

---

> > ### Comment · Reviewer_6hRL · 2023-10-24
> > **Clarifications**
> >
> > Hi, in this comment I'll respond to the clarification questions bolded in this part of the response. Thank you for your response!
> >
> > > Let us know if this is missing the concern here.
> >
> > So in (b), the main point is that finding a niche application is to thoroughly study multi-agent communication (as a subproblem within EL)?
> >
> > > By this, are you asking whether or not work in simplified environments is ultimately informative to richer environments?
> >
> > Yes, I think this is my question. My concern is that working in simplified environments might introduce underlying assumptions that don't generalize in more complex environments. (I totally agree this is an open question. We necessarily have to scope our work, and this often means simplifying the problem via simplifying the environment. My own work uses such environments. But, if there is a thought on _how_ we might measure the introduction of these biases and avoid developing solutions reliant on them, that would be nice to add in the paper.)
> >
> > > could you point out where this distinction needs to be made?
> >
> > Honestly I was asking more as a philosophical question; I don't think it absolutely needs to be made in this paper.
> >
> > > This is a topic we will need to look into more and address accordingly in the paper. Do you have any specific literature in mind?
> >
> > Nothing that I am intimately familiar with, unfortunately. I don't think it's critical to address in your paper. One paper that might be relevant to this is Smith et al. 2003, in ACS, "Complex systems in language evolution: The cultural emergence of compositional structure".

---

> ### Author Response · Authors · 2023-10-14
> **Reply (4/4)**
>
> (Select "Sort: Oldest First" if Reply 4/4 is first.)
>
> > How is AlphaGo analogous to generating synthetic language data?
>
> AlphaGo Zero (specifically) discovers strategies for playing Go via self-play,
> learning almost exclusively from environmental dynamics instead.  Emergent
> language discovers strategies for communicating and coordinating among agents,
> learning primarily from environmental dynamics.  Additionally, AGZ
> "rediscovered" many strategies familiar to human players of Go; analogously,
> the hope is that emergent language would also "rediscover" features of human
> language from functional advantages vis-a-vis the environment. In fact, you may
> say that AGZ generates synthetic Go matches in the same way that EL simulations
> generate utterances in synthetic languages.
>
>
> > In 3.1 Next steps: "different relative merits" what are these merits?
>
> Namely, Li et al.'s (2020) method of pretraining takes better advantage of the
> embodied training of the EL agent model while Yao et al.'s (2022) method
> decouples the EL agent's model from the target model of transfer, granting more
> flexibility in process.  Further explanations will be added to the paper.
>
>
> > 4.1 -- why is Current state and Next steps missing?
>
> Sec 4.1 is a "meta-section", stating the common features within "Description"
> and "Applicability" to avoid excessive repetition in subsequent sections.  For
> "Current state" and "Next steps", there is not so much commonality as to
> necessitate an entry in Sec 4.1.
>
>
> > What is "true multi-modal integration" (Section 4.2) and what would it look
> > like to have this true integration?
>
> By integration we mean embodiment and interaction in the way the we would
> expect a child to acquire language (along the lines of the arguments in
> "Experience Grounds Language" (Bisk et al., 2020)).
>
>
> > How could we know if our findings in EL actually generalize to human language
> > change? (Section 4.4)
>
> There would need to be concordance with historical examples of language change
> as well as current linguistic theories of language change. There have been
> almost two hundred years of scientific research into language change and, in
> this time, a great deal has been learned about how languages change (especially
> in terms of empirical observations). We can say that EL generalizes to human
> language change if the processes observed in EL simulations account to
> a non-trivial degree for the facts observed by historical linguists. For
> example, based on historical linguistic observations, we expect units of
> language to start as concrete signs that refer to specific classes of things in
> the environment. We expect more abstract, general units to develop, over
> generations, from these more concrete units.
>
> > Why does section 4.5 refer to non-linguistic perception/action/etc. as
> > "pre-linguistic inputs" rather than "non-linguistic inputs"?
>
> Children generally acquire a lot of social, perceptual, motor, etc. skills
> before language understanding and production, so "pre-linguistic" emphasizes
> the foundational nature of those non-linguistic inputs.  Nevertheless, this
> might be too specific, and we could use "non-linguistic" instead.
>
>
> ## Broader Impact Concerns:
>
> > I would like somewhat more discussion on broader impacts wrt. generalization
> > of EL to the significant variation in human language use across and within
> > language communities. There is a bit of discussion on low-resource languages,
> > but a clear discussion of how existing work engages with existing human
> > languages (e.g., what languages do they evaluate against?) and variance
> > across individual language agents within a community is not discussed at all.
>
> The vast majority of existing EL research does not interact directly with particular
> human languages and rather deals with human language in the abstract.  To my
> knowledge, most of EL research is produced at European and North American
> universities which would likely give an Indo-European bias to notions of
> language in the abstract, but this is a bit harder to demonstrate beyond
> speculation.  __In light of this fact, can you clarify what you would like added?__
>
>
> Once again, thank you for excellent, thorough review.

---

> > ### Comment · Reviewer_6hRL · 2023-10-24
> > **Clarifications**
> >
> > > In light of this fact, can you clarify what you would like added?
> >
> > I'm not aware of work that fits under this umbrella -- all I was suggesting is that since there is some discussion on applications of EL to low-resource languages (in section 3.1), I was assuming this is something that has been published on (to some extent). Perhaps my confusion is coming from "relevant two papers" in the first paragraph of "Current state" of 3.1 not being explicitly cited in-line. Though I suppose here "low-resource" doesn't necessarily mean these two studies were evaluating on low-resource human languages.
> >
> > Thank you so much for your response!

---

### Review · Reviewer_Tip2 · 2023-10-09

**Summary Of Contributions:**

This paper provides a thorough and informative review of research on deep-learning approaches to emergent language. The paper covers core research in emergent language, as well as potential applications and relevance to the cognitive science of language.

**Audience:**

Yes

**Broader Impact Concerns:**

There are not any discernible ethical implications arising from this work.

**Claims And Evidence:**

Yes

**Requested Changes:**

The authors may wish to address some of the suggestions mentioned above.

**Strengths And Weaknesses:**

I found this paper to be a well-written, thorough, and highly informative review of deep-learning approaches to emergent language. I think this will be an impactful contribution to the literature, collecting and organizing a wide range of related research, and helping to clarify promising directions for future research. I have some suggestions below that may help to further improve the paper:

- Although it is helpful to have an explicit framework for organizing the paper, the framework did not seem to fit the material in some sections. For example, organizing the paper around different applications was a bit confusing, given that 'intermediate applications' (more like basic research on emergent language) aren't applications at all. Similarly, the section on 'Rederiving human language' is treated as one research area among many, but it seems that it would be better described as the overarching goal of basic research in emergent language, while the following subsections (metrics, theoretical models, tools) are means for achieving that goal. I think it may make more sense not to impose the same structure on each of the sections, given the very different nature of basic research and applications.
- It might be good to include a few figures or boxes illustrating some notable or canonical examples of deep-learning-based emergent language research. This would be helpful particularly to readers who are less familiar with the topic.
- I found the contribution of emergent language research to language acquisition to be somewhat unclear. The primary source of confusion is that language acquisition typically occurs in the context of pre-existing languages. For the research described in this section, is the emergent nature of the language somehow important to the conclusions derived from the research, or is it merely incidental? The emphasis seems to be on the fact that emergent language research typically involves more realistic settings (embodiment, multimodality), and these are likely important factors in human language acquisition, but wouldn't it be equally possible to incorporate these factors in the study of language acquisition without the need for emergent language per se?

Minor notes:
- I was confused by the following statement in section 3.2: '…although these comparisons use the competitors as baselines rather than comparing them head-to-head to show the real-world superiority of emergent language-based communication.’ What is the distinction between a comparison to baselines and a 'head-to-head' comparison?
- There are a number of sentence with missing words, or other similar errors, throughout the paper.

---

> ### Author Response · Authors · 2023-10-14
> **Reply**
>
> Thank you for the review.  We feel that the review has assessed the paper well,
> and we plan to revise the paper in light of the comments.  We offer our
> specific responses and clarification questions below.
>
>
> > Although it is helpful to have an explicit framework for organizing the
> > paper, the framework did not seem to fit the material in some sections. ...
>
> Yes, this was a challenge of writing the "Intermediate Applications" section (to
> keep it consistent without shoehorning).  __Do you think, then, that the content
> of section is largely appropriate in its current state but that we should
> change the headings/organization to more naturally fit the structure and
> terminology of basic research?__  Or do you think that more extensive revision of
> the section is needed in order to appropriately convey the nature of
> "rederiving human language" and the rest of the basic research topics.
>
>
> > It might be good to include a few figures or boxes illustrating...
>
> We agree entirely and will do so.
>
>
> > I found the contribution of emergent language research to language
> > acquisition to be somewhat unclear. The primary source of confusion is that
> > language acquisition typically occurs in the context of pre-existing
> > languages. For the research described in this section, is the emergent nature
> > of the language somehow important to the conclusions derived from the
> > research, or is it merely incidental? The emphasis seems to be on the fact
> > that emergent language research typically involves more realistic settings
> > (embodiment, multimodality), and these are likely important factors in human
> > language acquisition, but wouldn't it be equally possible to incorporate
> > these factors in the study of language acquisition without the need for
> > emergent language per se?
>
> Yes, in general, it is possible to use a human language to train neural models
> in order to investigate language acquisition, and this can be done in
> a realistic (i.e., embodied, interactive) setting.  What emergent language
> offers beyond this is that the acquisition process we can observe is the real
> acquisition process for the language.  This would be the case in an EL which is
> formed through successive generations of new agents being "born" into the
> population and old agents "dying out".  When modelling the acquisition of
> language with a human language, we are only approximating how that human
> language is acquired since human language came about by different
> acquisition/transmission processes.
>
>
> > I was confused by the following statement in section 3.2: '…although these
> > comparisons use the competitors as baselines rather than comparing them
> > head-to-head to show the real-world superiority of emergent language-based
> > communication.’ What is the distinction between a comparison to baselines and
> > a 'head-to-head' comparison?
>
> By "baseline" we mean using an approach to give a reference point for the performance of a reasonable yet simple method.  By "head-to-head" we mean that the alternative method is competitive and presented in its strongest form, not just as a point of reference.  A head-to-head comparison is more focused on establishing practical, real-world superiority and not just giving a point of reference or sanity check.  For example, if you were developing a new contextual embedding model, comparing against word2vec (the "baseline") might give you general idea of how well it is doing, but you would need to compare against, say, BERT to determine if your method is of practical interest.
>
> We will clarify this in the paper.

---

> > ### Comment · Reviewer_Tip2 · 2023-10-21
> > **Response**
> >
> > Thanks very much to the authors for this reply. My responses are below:
> >
> > - I don't think a substantial reorganization is needed. I think it would suffice to change the headings, and potentially remove/compress some of the content (e.g. the subsection on the 'applicability' rederiving human language doesn't seem necessary, since it's already apparent why this is essential to emergent language research). This is just a suggestion however.
> > - I take the authors point about multi-generational emergent language, and I see how the generations that follow the original emergence of the language can serve as a good model of language acquisition. But it's still not clear to me why this is essential for studying language acquisition. The paper emphasizes things like embodiment and interactivity, but these seems like features that could be integrated into language acquisition research without the need for the language to be emergent (e.g. with pre-existing human languages). So, I don't doubt that emergent language research can shed light on language acquisition, but it would be helpful if the authors could clarify whether it makes any *unique* contributions to this research question.
> > - I now understand the distinction re: 'baselines', thanks for the clarification.

---

### Review · Reviewer_uFsU · 2023-10-09

**Summary Of Contributions:**

This review paper offers a survey of the field of deep learning-based emergent communication or emergent language, where neural network agents are trained to communicate with each other in simulated multi-agent environments, and the resulting communication protocols are then used, either as a testbed for cognitive or linguistic questions around language emergence and use, or more directly for downstream NLP tasks. The authors categorize this work into three sections: first, basic advances in improving the quality and capabilities of EC systems; second, the use of EC for downstream tasks relevant for NLP and ML; and finally, the use of EC to answer scientific questions on the origin and use of language in humans.

The authors outline much of the current state-of-the-art in the area, and in each section conclude with outstanding challenges for the field.

**Audience:**

Yes

**Claims And Evidence:**

Yes

**Requested Changes:**

# Too many walls of text; more figures and examples throughout.

My main concern is that the paper could include more concrete examples and figures in many sections. There are basically no figures or examples between the first and last sections of the paper. Most of the intermediate sections are just “walls of text” with many citations and a cursory description of certain papers. This is bad for readability, but the lack of examples also means that a reader might not be able to understand how these papers work in detail, and hence understand what might be missing from the current SoTA.

For example, “metrics for compositionality” (2.1) is one of the most important aspects in current EC research, but most metrics are described relatively briefly and none are explained in detail. Understanding the metrics more clearly will help the reader understand what the current limitations are of the metrics. It would be very clear upon understanding metrics, such as topographic similarity (a very common one), that there are clear limitations of current measures of compositionality—perhaps authors would then be able to make a more precise recommendation as to what is missing from the current set of metrics, rather than “we do not present any general recommendations.” I believe such section should also cite papers that reveal no clear link between current measures of compositionality and task performance, which helps the reader understand the current state of metrics for the field (https://aclanthology.org/2020.acl-main.407/)

Another example, “theoretical models” (2.3): it’s hard to understand what a concrete “theoretical model” looks like from just reading section 2.3. A concrete toy example of a theoretical model, either crafted by the authors, or simply taken from an existing paper, would go a long way in understanding this section.

# Vague next steps

I think as a result of the relatively cursory description of these sections, a lot of the sections end with “There are no general ‘next steps’ here.” I’m not sure how accurate this is—I think by more clearly describing the various sections and their limitations, it would be clearer what the next steps are for the field as a whole.

For example in 2.1 “rederiving natural language”: authors state that next steps could include “identifying the most salient properties of human language and using these to develop a concrete problem definition of “rederiving human language.” Linguistics would be the primary resource for identifying such properties from a theoretical standpoint.” But not much more is said. While obviously the critical components of human language, and the specific aspects of such an agenda, are up for debate, the authors could better summarize what aspects are currently missing from the current EC literature if we were to propose a “rederiving human language” thrust (e.g. model scale, population size, embodiment, task realism, any linguistic constraints, etc)

# Currently presumes familiarity with EC

Related to the above discussion on how there are insufficient examples throughout, this also translates into the introduction, where I believe the paper wouldn’t be easily understandable to someone without existing familiarity with EC. The  paper could set up a basic EC-style setup before going deep into applications/goals of sections 2, 3, and 4. Sketching what a typical EC setup looks like in Section 1 (perhaps around when the typical/necessary components of EC are sketched out) would be enormously helpful in situating the reader. For example consider Figure 3 in Lazaridou and Baroni (2020) which outlines a basic EC setup.

# Should make a clearer distinction from other reviews of EC

For example, the distinction between the Lazaridou and Baroni (2020) paper in Section 1.2 is a bit unclear. Lazaridou and Baroni is quoted as “presenting a much broader view of emergent language”, while this review is focused on “the goals and applications of emergent language research.”. This is a bit unclear to me—I would say the Lazaridou and Baroni paper is certainly interested in the goals/applications of emergent language research. It’s also possible that this review itself is broader/more up-to-date since it’s longer and has been released more recently. But I think any clarification of differences between this paper and others would be valuable (why do we need both?)

# Section 4 specific: differences between non-deep-RL approaches

(This is not as crucial of a blocker/recommendation, but something to consider)


Computer simulations have been used for a very long time to try to simulate and understand certain properties of language use and emergence. I think the authors are aware of this—each individual subsection in Section 4 describes why deep learning might particularly be useful for understand e.g. language acquisition (since it’s currently the most “realistic” simulation we can set up)—but the overall effect is that the main reason(s) why practitioners might reach for DL tools when answering linguistic questions is buried/weaved throughout the various sections here.

I would consider adding a specific section to Section 4 that explicitly describes why one might use DL methods to answer these linguistic questions. There are clear benefits, but also clear weaknesses (despite our best efforts, agents training in simulation don’t acquire anything close to human languages, and there are many outstanding artificialities in our current training paradigm). A centralized place to discuss the benefits/weaknesses of DL, and how DL fills in gaps left by the pre-deep-RL literature, would make Section 4 clearer.

# Summary

In summary I believe there was clearly a lot of effort put into this paper, and I believe it could make a useful contribution to the literature, but I would say it is not quite ready for publication just yet. The main blocker of my recommendation is that the middle is some ~20 continuous pages of text, and even after reading all of the text I don’t think the reader has the clearest sense of the work in some sections, due to a lack of figures or more concrete examples/explanations of some of the phenomena described in the text (e.g. lack of concrete discussion of metrics for emergent languages and their weaknesses, or key “results” figures from relevant papers in sections 3 and 4).

Therefore, to secure my recommendation, I would likely want
-  (**most crucial**) Improvements in writing and paper structure in the main body of the text, with additional figures + examples, and more pointed explanations for what is lacking from current research;
- (**nice to have**) some additional pedagogical context for new readers,
- (**nice to have**) more context as to why we need this specific review of the EC literature, and not existing reviews (most notably Lazaridou & Baroni 2020, which is the closest paper to this one).

# Other minor suggestions

- I prefer “emergent communication” over “emergent language”—emergent language is the thing being studied but I find “emergent communication” to more commonly refer to the subfield itself
- I think the de novo point in the abstract can be misleading since not all EC work studies emergence from scratch; there is plenty of work that uses EC-based techniques to augment existing systems trained with human languages, e.g. https://arxiv.org/abs/2203.13344 and https://arxiv.org/abs/1909.04499, much of which the authors cite
- I find “intermediate applications” a bit of a confusing word, since it’s not clear what the reference class is when saying something is “intermediate.” Case in point, the authors themselves claim that “Intermediate applications are not what we would typically consider applications at all.” It seems like a more appropriate term is something like basic/intrinsic advances in EC. I would consider rewording this (though I don’t feel very strongly either way)
- page 6: “No work in the current body of literature has explicitly pursued the rederivation of human language.” It’s worth clarifying in this sentence why this is (ie that the full “rederivation of human language” is an extremely ambitious goal with too many moving pieces)
- section 3.1 applicability: “Emergent language serves as a way to generate synthetic language data which more closely mimics the natural variation found in human language.” Is this true, given that existing studies show that such synthetic languages often look drastically different from human languages?

**Strengths And Weaknesses:**

# Strengths

- This paper is quite comprehensive and up-to-date. It could be a very useful reference for readers looking for relevant papers on EC.
- Clear thought and care was put into organizing the paper across different applications (sections 2, 3, and 4), as well as presenting the current SoTA and missing work in each section (with the description/applicability/current state/next steps) framework. As a result, the paper is systematic and relatively easy to read.

---

> ### Author Response · Authors · 2023-10-14
> **Reply (1/2)**
>
> Clarification questions for the reviewer are __bolded__ below.
>
>
> ## Re: Too many walls of text; more figures and examples throughout.
>
> > My main concern is that the paper could include more concrete examples and
> > figures in many sections. There are basically no figures or examples between
> > the first and last sections of the paper. Most of the intermediate sections are
> > just “walls of text” with many citations and a cursory description of certain
> > papers. This is bad for readability, but the lack of examples also means that
> > a reader might not be able to understand how these papers work in detail, and
> > hence understand what might be missing from the current SoTA.
>
> We acknowledge the paper will be more readable once we add:
> 1. More concrete examples
> 2. More illustrations
> We feel that the many references are necessary--this is a review paper--but we
> agree that adding more concrete detail about and illustration of SotA
> approaches would greatly strengthen the paper. We can definitely address this
> weakness and will make it a top priority.
>
> > For example, "metrics for compositionality" (2.1) is one of the most
> > important aspects in current EC research, but most metrics are described
> > relatively briefly and none are explained in detail.
>
> We could expand this section, although it is already one of the longest
> sections in the paper.  One way to address this, then could to be explain
> a couple of the most salient metrics (like topographic similarity for
> compositionality) without giving a full explanation of every single metric.
> __Would you find this approach to be acceptable?__
>
>
> > I believe such section should also cite papers that reveal no clear link
> > between current measures of compositionality and task performance, which
> > helps the reader understand the current state of metrics for the field
> > (https://aclanthology.org/2020.acl-main.407/)
>
> Thank you for this suggestion. We will add this paper to this section.
>
>
> > Another example, “theoretical models” (2.3) ... A concrete toy example of
> > a theoretical model [should be added].
>
> We have two examples in mind and will add one.
>
>
> ## Re: Vague next steps
>
> > I think as a result of the relatively cursory description of these sections,
> > a lot of the sections end with “There are no general ‘next steps’ here.” ...
>
> We will replace these vague "Next steps" sections with more specific
> recommendations.
>
>
> > For example in 2.1 "rederiving natural language" ... the authors could better
> > summarize what aspects are currently missing from the current EC literature
> > if we were to propose a “rederiving human language” thrust (e.g.  model
> > scale, population size, embodiment, task realism, any linguistic constraints,
> > etc)
>
> We will also specifically address this concern.
>
>
> ## Re: Currently presumes familiarity with EC
>
> > Related to the above discussion on how there are insufficient examples
> > throughout, ... The paper could set up a basic EC-style setup ... Sketching
> > what a typical EC setup looks like in Section 1
>
> We will add this as well.
>
>
> ## Re: Should make a clearer distinction from other reviews of EC
>
> > For example, the distinction between the Lazaridou and Baroni (2020) paper in
> > Section 1.2 is a bit unclear. ... I think any clarification of differences
> > between this paper and others would be valuable (why do we need both?)
>
> While Lazaridou & Baroni (2020) do briefly mention some of the goals of
> emergent communication, they do so as one small part of many facets of emergent
> language.  This paper intends to give a detailed, systematic, and comprehensive
> review of the goals specifically and not the other facets of emergent language.
> While the former paper might spend 1-2 pages explicating the goals of emergent
> communication, the whole of this paper is dedicated to it.  We will express this more clearly in the paper.

---

> ### Author Response · Authors · 2023-10-14
> **Reply (2/2)**
>
> ## Re: Section 4 specific: differences between non-deep-RL approaches
>
> > I would consider adding a specific section to Section 4 that explicitly
> > describes why one might use DL methods to answer these linguistic questions.
> > There are clear benefits, but also clear weaknesses (despite our best
> > efforts, agents training in simulation don’t acquire anything close to human
> > languages, and there are many outstanding artificialities in our current
> > training paradigm). A centralized place to discuss the benefits/weaknesses of
> > DL, and how DL fills in gaps left by the pre-deep-RL literature, would make
> > Section 4 clearer.
>
> This what we attempted (perhaps unsuccessfully) to do in Sec 4.1 ("General
> paradigm of knowledge-driven applications").  __Would you suggest that this
> section needs to elaborate further on the pros and cons DL or that DL itself,
> apart from EC, should be expanded upon in its own section?__
>
>
>
> ## Re: Summary
>
> > key "results" figures from relevant papers in sections 3 and 4
>
> __We are not quite sure what the "key 'results' figures" refers to.  Could you
> give some specific examples?__
>
>
>
> ## Re: Other minor suggestions
>
> > I prefer “emergent communication” over “emergent language”—emergent language
> > is the thing being studied but I find “emergent communication” to more
> > commonly refer to the subfield itself
>
> We are similarly aware that the field tends to favor "emergent communication"
> over "emergent language", so we are flexible on this point.  Our particular
> rationale for using "emergent language" is that it emphasizes the type of
> communication that pursued in the discipline.
>
>
> > I think the de novo point in the abstract can be misleading since not all EC
> > work studies emergence from scratch; there is plenty of work that uses
> > EC-based techniques to augment existing systems trained with human languages,
> > e.g. https://arxiv.org/abs/2203.13344 and https://arxiv.org/abs/1909.04499,
> > much of which the authors cite
>
> While not every EC approach strictly studies "de novo" emergence, we believe it
> best characterizes the field of EC as a whole, distinguishing it from other
> areas of research like modelling historical language change directly from data,
> using RL and self-play to tune dialog agents, or studying emergent linguistic
> features in large language models. We are happy to loosen this definition.
>
>
> > I find “intermediate applications” a bit of a confusing word ...
>
> In light of this and other reviewers' comments, we will change the wording to
> something more straightforward.
>
>
> > section 3.1 applicability: “Emergent language serves as a way to generate
> > synthetic language data which more closely mimics the natural variation found
> > in human language.” Is this true, given that existing studies show that such
> > synthetic languages often look drastically different from human languages?
>
> We can edit sentence to emphasize the fact that EL is a _potential_ way to
> generate synthetic data, given appropriate design choices that push its
> structure towards that of human languages.
>
> Again, we thank your for your insightful and informed review!

---

> > ### Comment · Reviewer_uFsU · 2023-12-01
> > **Response**
> >
> > Thanks to authors for the detailed reply, and apologies for my lateness in my response:
> >
> > > We could expand this section, although it is already one of the longest sections in the paper. One way to address this, then could to be explain a couple of the most salient metrics (like topographic similarity for compositionality) without giving a full explanation of every single metric. Would you find this approach to be acceptable?
> >
> > Yes that seems good—I understand the concern to be concise in this section, but on the other hand think a brief description of some of the relevant measures would go a long way.
> >
> > > This what we attempted (perhaps unsuccessfully) to do in Sec 4.1 ("General paradigm of knowledge-driven applications"). Would you suggest that this section needs to elaborate further on the pros and cons DL or that DL itself, apart from EC, should be expanded upon in its own section?
> >
> > Yes, I think this is correct; it's not a crucial thing to fix, but it would be nice to have more discussion about why DL is uniquely suited to solve these problems (besides non-DL approaches to EC). It may or may not need its own section, I don't have a very strong opinion here.
> >
> > > We are not quite sure what the "key 'results' figures" refers to. Could you give some specific examples?
> >
> > I was thinking authors could include something along the lines of Figure 4 in https://arxiv.org/abs/2006.02419: that is, for certain key empirical results, graphics can help make it very clear what the relevant conclusions are. This helps for readers skimming the paper, since otherwise the key conclusions and findings are buried within walls of text. This is not the only way to improve readability but it would be a welcome one.
> >
> > I appreciate the authors addressing of my other concerns and the promised changes. This might take some non-significant revision, but assuming those revisions are done satisfactorily, I would be happy to see the paper accepted.

---

### Decision · Action_Editor_aTFx · 2023-12-13

**Recommendation:** Accept with minor revision

**Comment:**

This work provides a broad and up-to-date overview of emergent language / emergent communication research in the deep learning community. The general organization of related work would be quite useful for newcomers to the area.

The manuscript was reviewed by three expert reviewers in the area and all three recommend acceptance. During the review process, reviewers had many detailed requests for revisions and improvements to the exposition. Authors replied to these concerns -- often agreeing with the sentiments and committing to making the corresponding edits.

As this is a survey paper and thus providing clear exposition of the area is its goal -- the AE deems these edits to be important enough to warrant a minor revision prior to acceptance.

**Audience:**

All reviewers have assessed there to be an audience for this work at TMLR and the AE agrees. As a survey paper, I expect there are many people within (or interested in joining) the emergent language research community that might find it relevant.

**Claims And Evidence:**

As a survey paper, no experimental claims are made. Claims regarding the emergent language literature seem well-founded and three knowledgeable reviewers agree with this recommendation. Reviewer 6hRL raised several general claims made in the manuscript that should be altered, clarified, or otherwise be better supported. The author discussion clarified these and the modification are expected in a camera ready.

---

> ### Author Response · Authors · 2024-02-01
> **Camera-ready revisions**
>
> In the camera-ready version we have:
> - Addressed comments about issues with clarity, claims, citations, and typos.
> - Added numerous figures.
> - Expanded "Current State" in sections which did not already have concrete examples.
> - Provided more concrete recommendations in "Next Steps" sections.